# DOPE: Doubly Optimistic and Pessimistic Exploration for Safe Reinforcement Learning

**Archana Bura, Aria HasanzadeZonuzy, Dileep Kalathil,**
**Srinivas Shakkottai, Jean-Francois Chamberland**
Department of Electrical and Computer Engineering, Texas A&M University
`{archanabura,azonuzy,dileep.kalathil,sshakkot,chmbrlnd}@tamu.edu`

## Abstract

Safe reinforcement learning is extremely challenging–not only must the agent explore an unknown environment, it must do so while ensuring no safety constraint violations. We formulate this safe reinforcement learning (RL) problem using the framework of a finite-horizon Constrained Markov Decision Process (CMDP) with an unknown transition probability function, where we model the safety requirements as constraints on the expected cumulative costs that must be satisfied during all episodes of learning. We propose a model-based safe RL algorithm that we call Doubly Optimistic and Pessimistic Exploration (DOPE), and show that it achieves an objective regret $\tilde{O}(|\mathcal{S}|\sqrt{|\mathcal{A}|K})$ without violating the safety constraints during learning, where $|\mathcal{S}|$ is the number of states, $|\mathcal{A}|$ is the number of actions, and $K$ is the number of learning episodes. Our key idea is to combine a reward bonus for exploration (optimism) with a conservative constraint (pessimism), in addition to the standard optimistic model-based exploration. DOPE is not only able to improve the objective regret bound, but also shows a significant empirical performance improvement as compared to earlier optimism-pessimism approaches.

## 1 Introduction

Constrained Markov Decision Processes (CMDPs) impose restrictions that pertain to resource or safety constraints of the system. For example, the average radiated power in a wireless communication system must be restricted due to user health and battery lifetime considerations or the frequency of braking or accelerating in an autonomous vehicle must be kept bounded to ensure passenger comfort. Since these systems have complex dynamics, a constrained reinforcement learning (CRL) approach is attractive for determining an optimal control policy. But how do we ensure that safety or resource availability constraints are not violated while learning such an optimal control policy?

Our goal is to develop a framework for safe exploration without constraint violation (with high probability) for solving CMDP problems where the model is unknown. While there has been much work on RL for both the MDP and the CMDP setting, ensuring *safe exploration* in the CRL setting has received less attention. The problem is challenging, since we do not allow constraint violation either during learning or deployment, while ensuring a low regret in terms of the optimal objective. Our aim is to explore a model-based approach in an episodic setting under which the model (the transition kernel of the CMDP) is empirically determined as samples from the system are gathered.

There has been much recent interest in model-based RL approaches to solving a constrained MDP, the most relevant of which we have summarized in Table 1. The setup is of a finite horizon episodic CMDP, with a state space of size $|\mathcal{S}|$, an action space of size $|\mathcal{A}|$ and a horizon of length $H$. Regret is measured over the first $K$ episodes of the algorithm. Both the attained objective and constraint satisfaction are computed in an expected sense for a given policy. Allowing constraint violations during learning means that the algorithm suffers both an objective regret and a constraint regret.

36th Conference on Neural Information Processing Systems (NeurIPS 2022).

Table 1: Comparison of safe RL algorithms for CMDPs. The algorithms are: OptCMDP, OptCMDP-Bonus [18], AlwaysSafe [36] and OptPessLP [28]. This table is presented for $K \geq \text{poly}(|S|, |A|, H)$, with polynomial terms independent of $K$ omitted. Expected regret results are in high probability.

| ALGORITHM | MODEL OPTIMISM | REWARD OPTIMISM | CONSTRAINT PESSIMISM | OBJECTIVE REGRET | CONSTRAINT REGRET | EMPIRICAL PERF. |
|---|---|---|---|---|---|---|
| OptCMDP | ✓ | ✗ | ✗ | $\tilde{O}(|\mathcal{S}|\sqrt{|\mathcal{A}|H^4 K})$ | $\tilde{O}(|\mathcal{S}|\sqrt{|\mathcal{A}|H^4 K})$ | - |
| OptCMDP-B | ✗ | ✓ | ✗ | $\tilde{O}(|\mathcal{S}|\sqrt{|\mathcal{A}|H^4 K})$ | $\tilde{O}(|\mathcal{S}|\sqrt{|\mathcal{A}|H^4 K})$ | - |
| AlwaysSafe | ✓ | ✗ | HEURISTIC | UNKNOWN | 0 | ✗ |
| OptPess-LP | ✗ | ✓ | ✓ | $\tilde{O}(\frac{H^3}{\bar{C}-\bar{C}_b}\sqrt{|\mathcal{S}|^3|\mathcal{A}|K})$ | 0 | ✗ |
| DOPE | ✓ | ✓ | ✓ | $\tilde{O}(\frac{H^3}{\bar{C}-\bar{C}_b}|\mathcal{S}|\sqrt{|\mathcal{A}|K})$ | 0 | ✓ |

Examples of algorithms following this method are OptCMDP, OptCMDP-Bonus [18]. The algorithms use different ways of incentivizing exploration to obtain samples to build up an empirical system model. OptCMDP uses the idea of optimism in the face of uncertainty from the model perspective and solves an extended linear program to find the best model (highest reward) under the samples gathered thus far. OptCMDP-Bonus uses a different approach of adding a bonus to the reward term to incentivize exploration of state-action pairs that have fewer samples, which we consider as a form of optimism from the reward perspective. While the objective regret of both these algorithms is $\tilde{O}(\sqrt{K})$, the constraint regret too is of the same order, since constraints may be violated during learning.

An algorithm that begins with no knowledge of the model will have exploration steps during learning that might violate constraints in expectation. Hence, safe RL approaches assume the availability of an inexpert baseline policy that does not violate the constraints, but is insufficient to explore the entire state action space. So it cannot be simply applied until a high-accuracy model is obtained. A heuristic approach entitled AlwaysSafe [36] assumes a factored CMDP that allows the easy generation of a safe baseline policy. It then combines the optimism with respect to the model of OptCMDP with a heuristically chosen hardening of constraints. This guarantees no constraint violations, but does not have regret guarantee with respect to the objective. Its empirical performance is variable and its use is limited to factored CMDP problems.

The OptPessLP algorithm [28] formalizes the idea of coupling optimism and pessimism by starting with OptCMDPBonus that has an optimistic reward, and systematically applying decreasing levels of pessimism with respect to the constraint violations. The approach is successful in ensuring the twin goals of a $\tilde{O}(\sqrt{K})$ objective regret, while ensuring no constraint violations. However, the authors do not present any empirical performance evaluation results. When we implemented OptPessLP, we found that the performance is singularly bad in that linear regret persists for a large number of samples, and the tapering off to $\tilde{O}(\sqrt{K})$ regret behavior does not appear to happen quickly. The problem with this algorithm is that it is so pessimistic with regard to constraints that it heavily disincentivizes exploration and ends up choosing the base policy for long sequences.

The issue upon which algorithm performance depends is the choice of how to combine optimism and pessimism to obtain both order optimal and empirically good objective regret performance, while ensuring no constraint violations happen. Our insight is that optimism with respect to the model is a key enabler of exploration, and can be coupled with the addition of optimism with respect to the reward. This *double dose of optimism*—both with respect to model and reward—could ensure that pessimistic hardening of constraints does not excessively retard exploration. Following this insight, we develop DOPE, a doubly optimistic and pessimistic exploration approach. We are able to show that DOPE not only attains $\tilde{O}(\sqrt{K})$ objective regret behavior with zero constraint regret with high probability, it also reduces the objective regret bound over OptPessLP by a factor of $\sqrt{|\mathcal{S}|}$. We conduct performance analysis simulations under representative CMDP problems and show that DOPE easily outperforms all the earlier approaches. Thus, the idea of double optimism is not only valuable from the order optimal algorithm design perspective, it also shows good empirical regret performance, indicating the feasibility of utilizing the methodology in real-world systems. The code for the experiments in this paper is located at: https://github.com/archanabura/DOPE-DoublyOptimisticPessimisticExploration

**Related Work: Constrained RL:** Constrained Markov Decision Processes (CMDP) has been an active area of research [2], with applications in domains such as power systems [39, 25], communication networks [3, 38], and robotics [17, 11]. In [9], the author proposed an actor-critic RL algorithm for

learning the asymptotically optimal policy for an infinite horizon average cost CMDP when the model is unknown. This approach is also utilized in function approximation settings with asymptotic local convergence guarantees [8, 12, 40]. Policy gradient algorithms for CMDPs have also been developed [1, 44, 47, 35, 16, 27, 49]. However, none these works address the problem of safe exploration to provide guarantees on the constraint violations during learning.

**Safe Multi-Armed Bandits:** The problem of safe exploration in linear bandits with stage-wise safety constraint is studied in [4, 24, 31, 33]. Linear bandits with more general constraints have also been studied [34, 29]. These works do not consider the more challenging RL setting which involves an underlying dynamical system with unknown model.

**Safe Online Convex Optimization:** Online convex optimization [20] has been studied with stochastic constraints [45, 10] and adversarial constraints [32, 46, 26]. These allow constraint violation during learning and characterize the cumulative amount of violation. A safe Frank-Wolfe algorithm for convex optimization with unknown linear stochastic constraints has been studied in [41]. However, these too do not consider the RL setting with an unknown model.

**Exploration in Constrained RL:** There has been much work in this area *with* constraint violations during learning, including the work discussed in the introduction [18]. These include [37, 19, 23], which derive bounds either on the objective and constraint regret or on the sample complexity of learning an $\epsilon$-optimal policy. Other works on safe RL include [43, 42], where a model-free approach is considered, and [5], that pertains to offline RL. These are complementary to our model-based approach. The problem of learning the optimal policy of a CMDP without violating the constraints was also studied in [48]. However, they assume that the model is known and only the cost functions are unknown, whereas we address more difficult problem with unknown model and cost functions.

**Notations:** For any integer $M$, $[M]$ denotes the set $\{1, \ldots, M\}$. For any two real numbers $a, b$, $a \vee b := \max\{a, b\}$. For any given set $\mathcal{X}$, $\Delta(\mathcal{X})$ denotes the probability simplex over the set $\mathcal{X}$, and $|\mathcal{X}|$ denotes the cardinality of the set $\mathcal{X}$.

## 2 Preliminaries and Problem Formulation

### 2.1 Constrained Markov Decision Process

We address the safe exploration problem using the framework of episodic Constrained Markov Decision Process (CMDP) [2]. We consider a CMDP, denoted as $M = \langle \mathcal{S}, \mathcal{A}, r, c, P, H, \bar{C} \rangle$ with $r = (r_h)_{h=1}^H, c = (c_h)_{h=1}^H, P = (P_h)_{h=1}^H$, where $\mathcal{S}$ is the state space, $\mathcal{A}$ is the action space, $H$ is the episode length, $r_h : \mathcal{S} \times \mathcal{A} \to \mathbb{R}$ is the objective cost function at time step $h \in [H]$, $c_h : \mathcal{S} \times \mathcal{A} \to \mathbb{R}$ is the constraint cost function at time step $h \in [H]$, $P_h$ is the transition probability function with $P_h(s'|s, a)$ representing the probability of transitioning to state $s'$ when action $a$ is taken at state $s$ at time $h$. In the RL context, the transition matrix $P$ is also called the model of the CMDP. Finally, $\bar{C}$ is a scalar that specifies the safety constraint in terms of the maximum permissible value for the expected cumulative constraint cost. We consider the setting where $|\mathcal{S}|$ and $|\mathcal{A}|$ are finite. Also, without loss of generality, we assume that costs $r$ and $c$ are bounded in $[0, 1]$.

A non-stationary randomized policy $\pi = (\pi_h)_{h=1}^H, \pi_h : \mathcal{S} \to \Delta(\mathcal{A})$ specifies the control action to be taken at each time step $h \in [H]$. In particular, $\pi_h(s, a)$ denotes the probability of taking action $a$ when the state is $s$ at time step $h$. For an arbitrary cost function $l : [H] \times \mathcal{S} \times \mathcal{A} \to \mathbb{R}$, the value function of a policy $\pi$ corresponding to time step $h \in [H]$ given a state $s \in \mathcal{S}$ is defined as

$$V_{l,h}^\pi(s; P) = \mathbb{E}[\sum_{\tau=h}^H l_\tau(s_\tau, a_\tau) | s_h = s], \tag{1}$$

where $a_\tau \sim \pi_\tau(s_\tau, \cdot)$, $s_{\tau+1} \sim P_\tau(\cdot|s_\tau, a_\tau)$. Since we are mainly interested in the value of a policy starting from $h = 1$, we simply denote $V_{l,1}^\pi(s; P)$ as $V_l^\pi(s; P)$. For the rest of the paper, we will assume that the initial state is $s_1$ is fixed. So, we will simply denote $V_l^\pi(s_1; P)$ as $V_l^\pi(P)$, when it is clear from the context. This standard assumption [18, 15] can be made without loss of generality.

The CMDP (planning) problem with a known model $P$ can then be stated as follows:
$$\min_\pi \quad V_r^\pi(P) \quad \text{s.t.} \quad V_c^\pi(P) \leq \bar{C}. \tag{2}$$
We say that a policy $\pi$ is a **safe policy** if $V_c^\pi(P) \leq \bar{C}$, i.e., if the expected cumulative constraint cost corresponding to the policy $\pi$ is less than the maximum permissible value $\bar{C}$. The **set of safe policies**,

denoted as $\Pi_{\text{safe}}$, is defined as $\Pi_{\text{safe}} = \{\pi : V_c^\pi(P) \leq \bar{C}\}$. Without loss of generality, we assume that the CMDP problem (2) is feasible, i.e., $\Pi_{\text{safe}}$ is non-empty. Let $\pi^*$ be the **optimal safe policy**, which is the solution of (2).

The CMDP (planning) problem is significantly different from the standard Markov Decision Process (MDP) (planning) problem [2]. Firstly, there may not exist an optimal deterministic policy for a CMDP, whereas the existence of a deterministic optimal policy is well known for a standard MDP. Secondly, there does not exist a Bellman optimality principle or Bellman equation for CMDP. So, the standard dynamic programming solution approaches which rely on the Bellman equation cannot be directly applied to solve the CMDP problem.

There are two standard approaches for solving the CMDP problem, namely the Lagrangian approach and the linear programming (LP) approach. Both approaches exploit the zero duality gap property of the CMDP problem [2] to find the optimal policy. In this work, we will use the LP approach, consistent with model optimism. Details of solving (2) using the LP approach are in Appendix A.

## 2.2 Reinforcement Learning with Safe Exploration

The goal of the reinforcement learning with safe exploration is to solve (2), but without the knowledge of the model $P$ a priori. Hence, the learning algorithm has to perform exploration by employing different policies to learn $P$. However, we also want the exploration for learning to have a safety guarantee, i.e, the policies employed during learning should belong to the set of safe policies $\Pi_{\text{safe}}$. Since $\Pi_{\text{safe}}$ itself is defined based on the unknown $P$, the learning algorithm will not know $\Pi_{\text{safe}}$ a priori. This makes the safe exploration problem extremely challenging.

We consider a model-based RL algorithm that interacts with the environment in an episodic manner. Let $\pi_k = (\pi_{h,k})_{h=1}^H$ be the policy employed by the algorithm in episode $k$. At each time step $h \in [H]$ in an episode $k$, the algorithm observes state $s_{h,k}$, selects action $a_{h,k} \sim \pi_{h,k}(s_{h,k}, \cdot)$, and incurs the costs $r_h(s_{h,k}, a_{h,k})$ and $c_h(s_{h,k}, a_{h,k})$. The next state $s_{h+1,k}$ is realized according to the probability vector $P(\cdot|s_{h,k}, a_{h,k})$. As stated before, for simplicity, we assume that the initial state is fixed for each episode $k \in [K] := \{1, \ldots, K\}$, i.e., $s_{1,k} = s_1$. We also assume that the maximum permissible cost $\bar{C}$ for any exploration policy is known and it is specified as part of the learning problem.

The performance of the RL algorithm is measured using the metric of *safe objective regret*. The safe objective regret is defined exactly as the standard regret of an RL algorithm for exploration in MDPs [21, 14, 6], but with an additional constraint that the exploration polices should belong to the safe set $\Pi_{\text{safe}}$. Formally, the safe objective regret $R(K)$ after $K$ learning episodes is defined as

$$R(K) = \sum_{k=1}^K (V_r^{\pi_k}(P) - V_r^{\pi^*}(P)), \ \pi_k \in \Pi_{\text{safe}}, \forall k \in [K]. \tag{3}$$

Since $\Pi_{\text{safe}}$ is unknown, clearly it is not possible to employ a safe policy without making any additional assumptions. We overcome this obvious limitation by assuming that the algorithm has access to a **safe baseline policy** $\pi_b$ such that $\pi_b \in \Pi_{\text{safe}}$. We formalize this assumption as follows.

**Assumption 1** (Safe baseline policy). The algorithm knows a safe baseline policy $\pi_b$ such that $V_c^{\pi_b}(P) = \bar{C}_b$, where $\bar{C}_b < \bar{C}$. The value $\bar{C}_b$ is also known to the algorithm.

*Remark* 2. Knowing a safe policy $\pi_b$ is necessary for solving the safe RL problem because we require the constraint to always be satisfied. A similar assumption has been used in the case of safe exploration in linear bandits [4, 24, 33], as well as in earlier work on safe RL [36, 28].

## 3 Algorithm and Performance Guarantee

DOPE builds on the *optimism in the face of uncertainty (OFU)* style exploration algorithms for RL [21, 14], using such optimism, both in terms of the model, as well as to provide a reward bonus for under-explored state-actions. However, a naive OFU-style algorithm may lead to selecting exploration policies that are not in the safe set $\Pi_{\text{safe}}$. So we modify the selection of exploratory policy by incorporating *pessimism in the face of uncertainty (PFU)* on the constraints, making DOPE doubly optimistic and pessimistic in exploration.

DOPE operates in episodes, each of length $H$. Define the filtration $\mathcal{F}_k$ as the sigma algebra generated by the observations until the end of episode $k \in [K]$, i.e., $\mathcal{F}_k = (s_{h,k'}, a_{h,k'}, h \in [H], k' \in [k])$. Let

$n_{h,k}(s,a) = \sum_{k'=1}^{k-1} \mathbb{1}\{s_{h,k'} = s, a_{h,k'} = a\}$ be the number of times the pair $(s,a)$ was observed at time step $h$ until the beginning of episode $k$. Similarly, define $n_{h,k}(s,a,s') = \sum_{k'=1}^{k-1} \mathbb{1}\{s_{h,k'} = s, a_{h,k'} = a, s_{h+1,k'} = s'\}$. At the beginning of each episode $k$, DOPE estimates the model as $\widehat{P}_{h,k}(s'|s,a) = n_{h,k}(s,a,s')/(n_{h,k}(s,a) \vee 1)$. Similar to OFU-style algorithms, we construct a confidence set $\mathcal{P}_k$ around $\widehat{P}_k$ as $\mathcal{P}_k = \cap_{(s,a) \in \mathcal{S} \times \mathcal{A}} \mathcal{P}_k(s,a)$, where

$$\mathcal{P}_k(s,a) = \{P' : |P'_h(s'|s,a) - \widehat{P}_{h,k}(s'|s,a)| \leq \beta^p_{h,k}(s,a,s'), \forall h \in [H], s' \in \mathcal{S}\}, \qquad (4)$$

$$\beta^p_{h,k}(s,a,s') = \sqrt{\frac{4\mathrm{Var}(\widehat{P}_{h,k}(s'|s,a))L}{n_{h,k}(s,a) \vee 1}} + \frac{14L}{3(n_{h,k}(s,a) \vee 1)}, \qquad (5)$$

where $L = \log(\frac{2SAHK}{\delta})$, and $\mathrm{Var}(\widehat{P}_{h,k}(s'|s,a)) = \widehat{P}_{h,k}(s'|s,a)(1 - \widehat{P}_{h,k}(s'|s,a))$. Using the empirical Bernstein inequality, we can show that the true model $P$ is an element of $\mathcal{P}_k$ for any $k \in [K]$ with probability at least $1 - 2\delta$ (see Appendix C).

Similarly, at the beginning of each episode $k$, DOPE estimates the unknown objective and constraint costs as $\hat{r}_{h,k}(s,a) = \frac{\sum_{k'=1}^{k-1} r_h(s,a)\mathbb{1}\{s_{h,k'}=s,a_{h,k'}=a\}}{n_{h,k}(s,a) \vee 1}$, $\hat{c}_{h,k}(s,a) = \frac{\sum_{k'=1}^{k-1} c_h(s,a)\mathbb{1}\{s_{h,k'}=s,a_{h,k'}=a\}}{n_{h,k}(s,a) \vee 1}$. In keeping with OFU, we construct confidence sets $\mathcal{R}_k$ and $\mathcal{C}_k$ around $\hat{r}_k$ and $\hat{c}_k$ respectively, as

$$\mathcal{R}_k = \{\tilde{r} : |\tilde{r}_h(s,a) - \hat{r}_{h,k}(s,a)| \leq \beta^l_{h,k}(s,a), \forall h, s, a \in [H] \times \mathcal{S} \times \mathcal{A}\},$$

$$\mathcal{C}_k = \{\tilde{c} : |\tilde{c}_h(s,a) - \hat{c}_{h,k}(s,a)| \leq \beta^l_{h,k}(s,a), \forall h, s, a \in [H] \times \mathcal{S} \times \mathcal{A}\}, \qquad (6)$$

$$\beta^l_{h,k}(s,a) = \sqrt{L'/(n^k_h(s,a) \vee 1)},$$

where $L' = 2\log(6SAHK/\delta)$, and $\tilde{r} = (\tilde{r}_h)_{h=1}^H, \tilde{c} = (\tilde{c}_h)_{h=1}^H$. Using Hoeffding inequality, we can show that the true costs belong to $\mathcal{R}_k$ and $\mathcal{C}_k$ for any $k \in [K]$ with probability at least $1 - \delta$ (see Appendix C). We define $\mathcal{M}_k = \mathcal{P}_k \cap \mathcal{R}_k \cap \mathcal{C}_k$ to be the total confidence ball.

It is tempting to use the standard OFU approach for selecting the exploration polices since this approach is known to provide sharp regret guarantees for exploration problems in RL. The standard OFU approach will find the optimistic model $\underline{P}_k$ and optimistic policy $\underline{\pi}_k$, where

$$(\underline{\pi}_k, \underline{P}_k) = \underset{\pi',(P',r',c') \in \mathcal{M}^k}{\arg\min} V^{\pi'}_{r'}(P'), \text{ s.t. } V^{\pi'}_{c'}(P') \leq \bar{C}. \qquad (7)$$

The OFU problem (7) is feasible since the true model $M$ is an element of $\mathcal{M}_k$ (with high probability). In particular, $(\pi_b, P)$ and $(\pi^*, P)$ are feasible solutions of (7). Moreover, (7) can be solved efficiently using an extended linear programming approach, as described in Appendix B. The policy $\underline{\pi}_k$ ensures exploration while satisfying the constraint $V^{\underline{\pi}_k}_c(\underline{P}_k) \leq \bar{C}$. However, this naive OFU approach overlooks the important issue that $\underline{\pi}_k$ may not be a safe policy with respect to the true model $P$. More precisely, it is possible to have $V^{\underline{\pi}_k}_c(P) > \bar{C}$ even though $V^{\underline{\pi}_k}_c(\underline{P}_k) \leq \bar{C}$. So, the standard OFU approach alone will not give a safe exploration strategy.

In order to ensure that the exploration policy employed at any episode is safe, we add a pessimistic penalty to the empirical constraint cost to get the pessimistic constraint cost function $\bar{c}_k$ as

$$\bar{c}_{h,k}(s,a) = \hat{c}_{h,k}(s,a) + \beta^l_{h,k}(s,a) + H\bar{\beta}^p_{h,k}(s,a), \qquad (8)$$

where $\bar{\beta}^p_{h,k}(s,a) = \sum_{s' \in \mathcal{S}} \beta^p_{h,k}(s,a,s')$. Since $\bar{\beta}^p_{h,k}(s,a)$ is $\tilde{O}(1/\sqrt{n_{h,k}(s,a)})$, $(s,a)$ pairs that are less observed have a higher penalty, disincentivizing their exploration. However, such a pessimistic penalty may prevent the exploration that is necessary to learn the optimal policy. To overcome this issue, we also modify the empirical objective cost function by subtracting a term to incentivize exploration, to obtain an optimistic objective cost function

$$\bar{r}_{h,k}(s,a) = \hat{r}_{h,k}(s,a) - \frac{3H}{\bar{C} - \bar{C}_b}\beta^l_{h,k}(s,a) - \frac{H^2}{\bar{C} - \bar{C}_b}\bar{\beta}^p_{h,k}(s,a). \qquad (9)$$

Since $\bar{\beta}^p_{h,k}(s,a)$ is $\tilde{O}(1/\sqrt{n_{h,k}(s,a)})$, $(s,a)$ pairs that are less observed will have a lowered cost to incentivize their exploration.

We select the policy $\pi_k$ for episode $k$ by solving the Doubly Optimistic-Pessimistic (DOP) problem:

$$(\pi_k, P_k) = \underset{\pi',P' \in \mathcal{P}_k}{\arg\min} V^{\pi'}_{\bar{r}_k}(P') \text{ s.t. } V^{\pi'}_{\bar{c}_k}(P') \leq \bar{C}. \qquad (10)$$

Notice that DOPE is doubly optimistic by considering both the optimistic objective cost function in (9) and the optimistic model $P_k$ from the confidence set $\mathcal{P}_k$ in (10), while being pessimistic on the

constraint in (10). Later, in Lemma 18 in the appendix, we prove that $(\pi_k, P_k)$ is indeed an optimistic solution. This is in contrast with [23], where the optimism is solely limited to the objective cost. We will show that our approach carefully balances double optimism and pessimism, yielding a regret minimizing learning algorithm with episodic safe exploration guarantees.

We note that the DOP problem (10) may not be feasible, especially in the first few episodes of learning. This is because, $\bar{\beta}^p_{h,k}(s,a)$ and $\bar{\beta}^l_{h,k}(s,a)$ may be large during the initial phase of learning so that there may not be a policy $\pi'$ and a model $P' \in \mathcal{P}_k$ that can satisfy the constraint $V^{\pi'}_{\bar{c}_k}(P') \leq \bar{C}$. We overcome this issue by employing a safe baseline policy $\pi_b$ (as defined in Assumption 1) in the first $K_o$ episodes, a value provided by Proposition 4. Since $\pi_b$ is safe by definition, DOPE ensures safety during the first $K_o$ episodes. We will later show that the DOP problem (10) will have a feasible solution after the first $K_o$ episodes (see Proposition 4). For any episode $k \geq K_o$, DOPE employs policy $\pi_k$, which is the solution of (10). We will also show that $\pi_k$ from (10) (once it becomes feasible) will indeed be a safe policy (see Proposition 5). We present DOPE formally in Algorithm 1.

---

**Algorithm 1** Doubly Optimistic and Pessimistic Exploration (DOPE)

---

1: **Input:** $\delta \in (0,1)$, $r, c, \pi_b, \bar{C}_b, \bar{C}, K_o$
2: **Initialization:** $n_{h,k}(s,a) = n_{h,k}(s,a,s') = 0$ $\forall s, s' \in S, a \in A, h \in [H]$.
3: **for** episodes $k = 1, \dots, K$ **do**
4:     Compute the estimates $\widehat{P}_k, \hat{r}_k, \hat{c}_k$ and the confidence set $\mathcal{M}_k$ according to (4) - (5), and (6).
5:     **if** $k \leq K_o$ **then**
6:         Select the exploration policy $\pi_k = \pi_b$
7:     **else**
8:         Select the exploration policy $\pi_k$ according to (10)
9:     **end if**
10:     **for** $h = 1, 2, \dots, H$ **do**
11:         Observe state $s_{h,k}$, select action $a_{h,k} \sim \pi_{h,k}(s_{h,k}, \cdot)$, incur the cost $r_h(s_{h,k}, a_{h,k})$ and $c_h(s_{h,k}, a_{h,k})$, and observe next state $s_{h+1,k} \sim P_h(\cdot|s_{h,k}, a_{h,k})$
12:         Update the counts: $n_{h,k}(s_{h,k}, a_{h,k}) \leftarrow n_{h,k}(s_{h,k}, a_{h,k}) + 1$, $n_{h,k}(s_{h,k}, a_{h,k}, s_{h+1,k}) \leftarrow n_{h,k}(s_{h,k}, a_{h,k}, s_{h+1,k}) + 1$
13:     **end for**
14: **end for**

---

We now present our main result, which shows that the DOPE algorithm achieves $\tilde{O}(\sqrt{K})$ regret *without* violating the safety constraints during learning, with high probability.

**Theorem 3.** *Fix any $\delta \in (0,1)$. Consider the DOPE algorithm with $K_o$ as specified in Proposition 4. Let $\{\pi_k, k \in [K]\}$ be the sequence of policies generated by the DOPE algorithm. Then, with probability at least $1 - 5\delta$, $\pi_k \in \Pi_{\text{safe}}$ for all $k \in [K]$. Moreover, with probability at least $1 - 5\delta$, the regret of the DOPE algorithm satisfies*

$$R(K) \leq \tilde{\mathcal{O}}(\frac{SH^3}{(\bar{C} - \bar{C}_b)}\sqrt{AK}).$$

## 4 Analysis

We now provide the technical analysis of DOPE, concluding with the proof outline of Theorem 3.

### 4.1 Preliminaries

For an arbitrary policy $\pi'$ and transition probability function $P'$, define $\epsilon^{\pi'}_k(P')$ and $\eta^{\pi'}_k(P')$ as

$$\epsilon^{\pi'}_k(P') = H\mathbb{E}[\sum_{h=1}^{H} \bar{\beta}^p_{h,k}(s_{h,k}, a_{h,k})|\pi', P', \mathcal{F}_{k-1}], \quad \eta^{\pi'}_k(P') = \mathbb{E}[\sum_{h=1}^{H} \beta^l_{h,k}(s_{h,k}, a_{h,k})|\pi', P', \mathcal{F}_{k-1}]. \tag{11}$$

Then, it is straightforward to show that (see (23) - (24) in the Appendix) $V^{\pi'}_{\bar{c}_k}(P') = V^{\pi'}_{\hat{c}_k}(P') + \eta^{\pi'}_k(P) + \epsilon^{\pi'}_k(P')$, $V^{\pi'}_{\bar{r}_k}(P') = V^{\pi'}_{\hat{r}_k}(P') - \frac{3H}{\bar{C}-\bar{C}_b}\eta^{\pi'}_k(P') - \frac{H}{\bar{C}-\bar{C}_b}\epsilon^{\pi'}_k(P')$. The analysis utilizes this decomposition of $V^{\pi'}_{\bar{c}_k}(P')$ and $V^{\pi'}_{\bar{r}_k}(P')$, and the properties of $\epsilon^{\pi'}_k(P')$ and $\eta^{\pi'}_k(P')$. Under the good event $G$ defined as in Lemma 12, we can show, $V^{\pi'}_{\hat{c}_k}(P') - \eta^{\pi'}_k(P') \leq V^{\pi'}_c(P') \leq V^{\pi'}_{\hat{c}_k}(P') + \eta^{\pi'}_k(P')$.

## 4.2 Feasibility of the OP Problem

Even though $(\pi_b, P)$ is a feasible solution to the original CMDP problem (2), it may not be a feasible for the DOP problem (10) in the initial phase of learning. To see this, note that $V_{\bar{c}_k}^{\pi_b}(P) = V_{\hat{c}_k}^{\pi_b}(P) + \eta_k^{\pi_b}(P) + \epsilon_k^{\pi_b}(P)$, and since $V_c^{\pi_b}(P) \geq V_{\hat{c}_k}^{\pi_b}(P) - \eta_k^{\pi_b}(P)$ under the good event, and $V_c^{\pi_b}(P) = \bar{C}_b$, we will have $V_{\bar{c}_k}^{\pi_b}(P) \leq \bar{C}$ if $2\eta_k^{\pi_b}(P) + \epsilon_k^{\pi_b}(P) \leq (\bar{C} - \bar{C}_b)$. So, $(\pi_b, P)$ is a feasible solution for (10) if $2\eta_k^{\pi_b}(P) + \epsilon_k^{\pi_b}(P) \leq (\bar{C} - \bar{C}_b)$. This sufficient condition may not be satisfied for initial episodes. However, since $\epsilon_k^{\pi_b}(P)$ and $\eta_k^{\pi_b}(P)$ are decreasing in $k$, if $(\pi_b, P)$ becomes a feasible solution for (10) at episode $k'$, then it will remain feasible for all episodes $k \geq k'$. Also, since $\bar{\beta}_{h,k}^p$ and $\beta_{h,k}^l$ decrease with $k$, one can expect that (10) becomes feasible after some number of episodes. We use these intuitions, along with some technical lemmas to show the following result.

**Proposition 4.** *Under the DOPE algorithm, with a probability greater that $1 - 5\delta$, $(\pi_b, P)$ is a feasible solution for the DOP problem (10) for all $k \geq K_o$, where $K_o = \tilde{\mathcal{O}}\left(\frac{S^2 A H^4}{(\bar{C} - \bar{C}_b)^2}\right)$.*

## 4.3 Safety Exploration Guarantee

We show that the DOPE algorithm provides a safe exploration guarantee, i.e., $\pi_k \in \Pi_{\text{safe}}$ for all $k \in [K]$ with high probability, where $\pi_k$ is the exploration policy employed by DOPE in episode $k$. This is achieved by the carefully designed pessimistic constraint of the DOP problem (10).

For any $k \leq K_o$, since $\pi_k = \pi_b$, and it is safe by Assumption 1. For $k \geq K_o$, (10) is feasible according to Proposition 4. Since $(\pi_k, P_k)$ is the solution of (10), we have $V_{\bar{c}_k}^{\pi_k}(P_k) = V_{\hat{c}_k}^{\pi_k}(P^k) + \eta_k^{\pi_k}(P_k) + \epsilon_k^{\pi_k}(P_k) \leq \bar{C}$. This implies that $V_{\hat{c}_k}^{\pi_k}(P_k) + \eta_k^{\pi_k}(P_k) \leq \bar{C} - \epsilon_k^{\pi_k}(P_k)$. We have that $V_c^{\pi_k}(P_k) \leq V_{\hat{c}_k}^{\pi_k}(P_k) + \eta_k^{\pi_k}(P_k)$ under the good event, and hence, the above equation implies that $V_c^{\pi_k}(P_k) \leq \bar{C} - \epsilon_k^{\pi_k}(P_k)$, i.e., $\pi_k$ satisfies a tighter constraint with respect to the model $P_k$. However, it is not obvious that the policy $\pi_k$ will be safe with respect to the true model $P$ because $V_c^{\pi_k}(P)$ may be larger than $V_c^{\pi_k}(P_k)$ due to the change from $P_k$ to $P$.

We, however, show that $V_c^{\pi_k}(P)$ cannot be larger than $V_c^{\pi_k}(P_k)$ by more than $\epsilon_k^{\pi_k}(P_k)$, i.e., $V_c^{\pi_k}(P) - V_c^{\pi_k}(P_k) \leq \epsilon_k^{\pi_k}(P_k)$. This will then yield that $V_c^{\pi_k}(P) \leq V_c^{\pi_k}(P_k) + \epsilon_k^{\pi_k}(P_k) \leq \bar{C}$, which is the true safety constraint. The key idea is in the design of the pessimistic cost function $\bar{c}_k(\cdot, \cdot)$ such that its pessimistic effect will balance the change in the value function (from $V_c^{\pi_k}(P_k)$ to $V_c^{\pi_k}(P)$) due to the optimistic selection of the model $P_k$. We formally state the safety guarantee of DOPE below.

**Proposition 5.** *Let $\{\pi_k, k \in [K]\}$ be the sequence of policies generated by the DOPE algorithm. Then $\pi_k$ is safe $\forall k \in [K]$, i.e., $V_c^{\pi_k}(P) \leq \bar{C}$, for all $k \in [K]$, with a probability greater than $1 - 5\delta$.*

## 4.4 Regret Analysis

The regret analysis for most OFU style RL algorithms follows the standard approach of decomposing the regret into two terms as $R_k = V_r^{\pi_k}(P) - V_r^{\pi^*}(P) = (V_r^{\pi_k}(P) - V_r^{\pi_k}(P_k)) + (V_r^{\pi_k}(P_k) - V_r^{\pi^*}(P))$, where $R_k$ denotes the regret in episode $k$. The first term is the difference between value functions of the selected policy $\pi_k$ with respect to the true model $P$ and optimistic model $P_k$. Bounding this term is the key technical analysis part of most of the OFU algorithms for the unconstrained MDP [21, 13] and also the CMDP [18]. In the standard OFU style analysis for the unconstrained problem, since $P \in \mathcal{P}_k$ for all $k$, it can be easily observed that $(\pi^*, P)$ is a feasible solution for the OFU problem (7) for all $k \in [K]$. Moreover, since $(\pi_k, P_k)$ is the optimal solution in $k^{th}$ episode, we get $V_r^{\pi_k}(P_k) \leq V_r^{\pi^*}(P)$. So, the second term will be non-positive, and hence can be dropped from the regret analysis. However, in our setting, the second term can be positive since $(\pi^*, P)$ may not be a feasible solution of the DOP problem (10) due to the pessimistic constraint. This necessitates a different approach for bounding the regret. Existing work [28] only considers optimism in the objective cost, and hence their proof closely follows that of OptCMDP-Bonus algorithm in [18] with pessimistic constraints. In analyzing the regret of DOPE, we need to handle the optimism in objective cost as well as the model in regret terms, along with the pessimistic constraints. This make the analysis particularly challenging. The full proof is detailed in the appendix.

# 5 Experiments

We now evaluate DOPE via experiments. We have two relevant metrics, namely, (i) objective regret, defined in (3) that measures the optimality gap of the algorithm, and (ii) constraint regret, defined

as $\sum_{k=1}^{K} \max\{0, V_c^{\pi_k}(P) - \bar{C}\}$, where $\pi_k$ is the output of the algorithm in question at episode $k$. This measures the safety gap of the algorithm. Our candidate algorithms are (i) OptCMDP, (ii) AlwaysSafe, (iii) OptPessLP and (iv) DOPE, all described in the introduction. OptCMDP is expected to show constraint violations, while the other three should show zero constraint regret. We consider two environments here, with a third environment presented in the appendix. AlwaysSafe can directly be used only with a factored CMDP, and only applies to the first environment presented. We simulate both variants of this algorithm, referred to as AlwaysSafe $\pi_T$ and AlwaysSafe $\pi_\alpha$, respectively [36].

**Factored CMDP:** We first consider a CMDP where the safety relavant features of the model can be separated, as shown in [36]. This CMDP has states $\{1, 2, 3\}$ arranged in a circle, and 2 actions $\{1, 2\}$ in each state, to move right or stay put, respectively. The transitions move the agent to its right state with probability 1, if action 1 is taken. If action 2 is taken, it remains in the same state with probability 1. Action 1 does not incur any objective cost or constraint cost. Action 2 incurs an objective cost equals to the state number, and a constraint cost of 1. We choose episode length $H = 6$, and constraint as $\bar{C} = 3$. The structure of this CMDP allows AlwaysSafe to extract a safe baseline policy from it.

**Media Streaming CMDP:** Our second environment represents media streaming to a device from a wireless base station, which provides high and low service rates at different costs. These service rates have independent Bernoulli distributions, with parameters $\mu_1 = 0.9$, and $\mu_2 = 0.1$, where $\mu_1$ corresponds to the fast service. Packets received at the device are stored in a media buffer and played out according to a Bernoulli process with parameter $\gamma$. We denote the number of incoming packets into the buffer as $A_h$, and the number of packets leaving the buffer $B_h$. The media buffer length is the state and evolves as $s_{h+1} = \min\{\max(0, s_h + A_h - B_h), N\}$, where $N = 20$ is the maximum buffer length. The action space is $\{1, 2\}$, where action 1 corresponds to using the fast service. The objective cost is $r(s, a) = \mathbb{1}\{s = 0\}$, while the constraint cost is $c(s, a) = \mathbb{1}\{a = 1\}$, i.e., we desire to minimize the outage cost, while limiting the usage of fast service. We consider episode length $H = 10$, and constraint $\bar{C} = \frac{H}{2}$.

**Experiment Setup:** OptCMDP and OptPessLP algorithms have not been implemented earlier. For accurate comparison, we simulate all the algorithms true to their original formulations of cost functions and confidence intervals. Our experiments are carried out for 20 random runs, and averaged to obtain the regret plots. For DOPE, we choose $K_0$ to be as specified in Proposition 4. Full details on the algorithm parameters and experiment settings are provided in Appendix E.

**Baseline Policies:** Both OptPessLP and DOPE require baseline policies. We select the baseline policies as the optimal solutions of the given CMDP with a constraint $\bar{C}_b = 0.2\bar{C}$. We choose the same baseline policies for both the algorithms. This choice is to showcase the efficacy of DOPE, despite starting with a conservative baseline policy.

**Results for Factored CMDP:** Fig. 1(a) shows the objective regret of the algorithms in this environment. OptCMDP has a good objective regret performance as expected, but shows constraint violations. AlwasySafe fails to achieve $\tilde{O}(\sqrt{K})$ regret in the episodes shown for both variants, although the $\pi_\alpha$ variant has smaller regret as compared to the $\pi_T$ variant. OptPessLP takes a long time to attain $\sqrt{K}$ behavior, which means that is chooses the baseline policy for an extended period, and shows high empirical regret. This suggests that reward optimism of OptPessLP is insufficient to balance the pessimism in the constraint. DOPE not only achieves the desired $\tilde{O}(\sqrt{K})$ regret, but also does so in fewer episodes compared to the other two algorithms. Furthermore, it has low empirical regret. Fig. 1(b) shows that the constraint violation regret is zero for all the episodes of learning for all the safe algorithms, while OptCMDP shows a large constraint violation regret.

**Results for Media Streaming CMDP:** Fig 2(a) compares the objective regret across algorithms. The value of DOPE over OptPessLP is more apparent here. After a linear growth phase, the objective regret of DOPE changes to a square-root scaling. OptPessLP has not explored sufficiently at this point, and hence suffers high linear regret. Finally, OptCMDP also has square-root regret scaling, but is fastest, since it is not constrained by safe exploration. Fig 2(b) compares the regret in constraint violation for these algorithms. As expected, DOPE and OptPessLP do not violate the constraint, while the OptCMDP algorithm significantly violates the constraints during learning.

**Effect of Baseline Policy:** We compare the objective regret of DOPE under different baseline policies in Fig 1(c) and Fig 2(c). We see that less conservative (but safe) baselines result in lower regret, but the difference is not excessive, implying that the exact baseline policy chosen is not crucial.

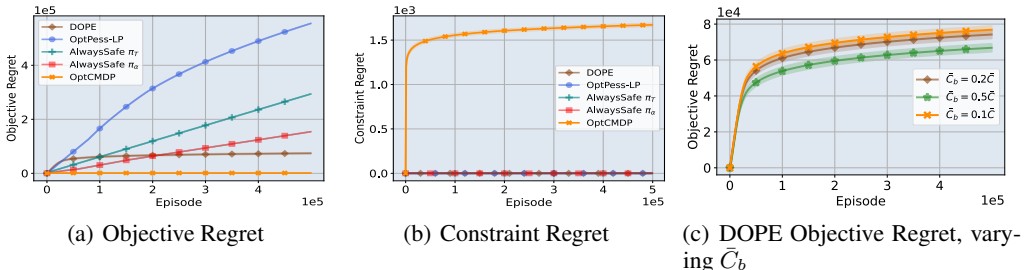

(a) Objective Regret     (b) Constraint Regret     (c) DOPE Objective Regret, vary-ing $\bar{C}_b$

Figure 1: Illustrating the Objective Regret and Constraint Regret for a Factored CMDP environment.

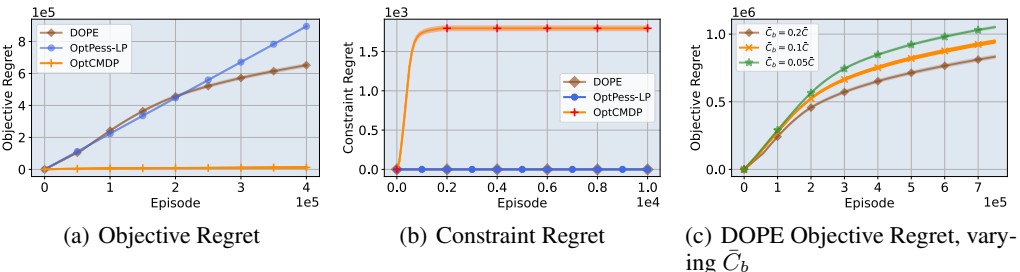

(a) Objective Regret     (b) Constraint Regret     (c) DOPE Objective Regret, vary-ing $\bar{C}_b$

Figure 2: Illustrating the Objective Regret and Constraint Regret for the Media Streaming Environment.

**Summary**: DOPE has two valuable properties: $(i)$ Faster rate of shift to $\sqrt{K}$ behavior, since the linear regret phase where it applies the baseline policy is relatively short, and $(ii)$ $\tilde{O}(\sqrt{K})$ regret with respect to optimal, which together mean that the empirical regret is lower than other approaches.

**Limitations:** Our goal is to find an RL algorithm with no constraint violation with high probability in the tabular setting. Since safe exploration is a basic problem, we follow the usual approach in the literature of first establishing the fundamental theory results for the tabular setting [21, 14, 6]. We also note that most of the existing work on exploration in safe RL is in the tabular setting [18, 37, 28]. In our future work, we plan to employ the DOPE approach in a function approximation setting.

# 6    Conclusion

We considered the safe exploration problem in reinforcement learning, wherein a safety constraint must be satisfied during learning and evaluation. Earlier approaches to constrained RL have proposed optimism on the model, optimism on reward, and pessimism on constraints as means of modulating exploration, but none have shown order optimal regret, no safety violation, and good empirical performance simultaneously. We started with the conjecture that double optimism combined with pessimism is the key to attaining the ideal balance for fast and safe exploration, and design DOPE that carefully combines these elements. We showed that DOPE not only attains order-optimal $\tilde{O}(\sqrt{K})$ regret without violating safety constraints, but also reduces the best known regret bound by a factor of $\sqrt{|\mathcal{S}|}$. Furthermore, it has significantly better empirical performance than existing approaches. We thus make a case for adoption of the approach for real world use cases and extension to large scale RL problems using function approximation.

# 7    Acknowledgement

This work was supported in part by the grants NSF-CAREER-EPCN 2045783, NSF ECCS 2038963, and ARO W911NF-19-1-0367. Any opinions, findings, and conclusions or recommendations expressed in this material are those of the authors and do not necessarily reflect the views of the sponsoring agencies.

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

## Societal Impact and Ethics Statement

Reinforcement learning has much potential for application to a variety of cyber-physical systems, such as the power grid, robotics and other systems where guarantees on the operating region of the system must be met. Our work provides a theoretical basis for the design of controllers that can be applied in such scenarios. The approaches presented in the paper were tested on simulated environments, and did not involve any human interaction. We do not see any ethical concerns with our research approach.

A note of caution with our approach is that the policy generated is only as good as the training environment, and many examples exist wherein the policy generated is optimal according to its training, but violates basic truths known to human operators and could fail quite badly. Indeed, our approach does not provide sample-path guarantees, and the system could well move into deleterious states for a small fraction of the time, which might be completely unacceptable and trigger hard fail safes, such as breakers in a power system. Understanding the right application environments with excellent domain knowledge is hence needed before any practical success can be claimed.

