# A Linear Programming Method for Solving the CMDP Problem

Here we give a brief description on solving the CMDP problem (2) using the linear programming method when the model $P$ is known. The details can be found in [18, Section 2].

The first step is to reformulate (2) using *occupancy measure* [2, 50]. For a given policy $\pi$ and an initial state $s_1$, the state-action occupation measure $w^\pi$ for the MDP with model $P$ is defined as

$$w_h^\pi(s,a;P) = \mathbb{E}[\mathbb{1}\{s_h = s, a_h = a\}|s_1, P, \pi] = \mathbb{P}(s_h = s, a_h = a|s_1, P, \pi). \tag{12}$$

Given the occupancy measure, the policy that generated it can easily be computed as

$$\pi_h(s,a) = \frac{w_h^\pi(s,a;P)}{\sum_b w_h^\pi(s,b;P)}. \tag{13}$$

The occupancy measure of any policy $\pi$ for an MDP with model $P$ should satisfy the following conditions. We omit the explicit dependence on $\pi$ and $P$ from the notation of $w$ for simplicity.

$$\sum_a w_h(s,a) = \sum_{s',a'} P(s|s',a')w_{h-1}(s',a'), \quad \forall h \in [H] \setminus \{1\} \tag{14}$$

$$\sum_a w_1(s,a) = \mathbb{1}\{s = s_1\}, \quad \forall s \in \mathcal{S}, w_h(s,a) \geq 0, \quad \forall(s,a,h) \in \mathcal{S} \times \mathcal{A} \times [H] \tag{15}$$

From the above conditions, it is easy to show that $\sum_{s,a} w_h(s,a) = 1$. So, occupancy measures are indeed probability measures. Since the set of occupancy measures for a model $P$, denoted as $\mathcal{W}(P)$, is defined by a set of affine constraints, it is straight forward to show that $\mathcal{W}(P)$ is convex. We state this fact formally below.

**Proposition 6.** *The set of occupancy measures for an MDP with model $P$, denoted as $\mathcal{W}(P)$, is convex.*

Recall that the value of a policy $\pi$ for an arbitrary cost function $l : \mathcal{S} \times \mathcal{A} \to \mathbb{R}$ with a given initial state $s_1$ is defined as $V_l^\pi(P) = \mathbb{E}[\sum_{h=1}^H l(s_h, a_h)|s_1 = s, \pi, P]$. It can then be expressed using the occupancy measure as

$$V_l^\pi(P) = \sum_{h,s,a} w_h^\pi(s,a;P)\, l_h(s,a) = l^\top w^\pi(P),$$

where $w^\pi(P) \in \mathbb{R}^{SAH}$ with $(s,a,h)$ element is given by $w_h^\pi(s,a;P)$ and $l \in \mathbb{R}^{SAH}$ with $(s,a,h)$ element is given by $l_h(s,a)$. The CMDP problem (2) can then be written as

$$\min_\pi r^\top w^\pi(P) \quad \text{s.t.} \quad c^\top w^\pi(P) \leq \bar{C}. \tag{16}$$

Using the properties of the occupancy measures, the reformulated CMDP problem (16) can be rewritten as an LP, where the optimization variables are occupancy measures [50, 18]. More precisely, the CMDP problem (2) and its equivalent (16) can be written as

$$\min_w \sum_{h,s,a} w_h(s,a)r_h(s,a) \tag{17a}$$

$$\text{subject to} \quad \sum_{h,s,a} w_h(s,a)c_h(s,a) \leq \bar{C} \tag{17b}$$

$$\sum_a w_h(s,a) = \sum_{s',a'} P_{h-1}(s|s',a')w_{h-1}(s',a'), \forall h \in [H] \setminus \{1\} \tag{17c}$$

$$\sum_a w_1(s,a) = \mathbb{1}\{s = s_1\}, \quad \forall s \in \mathcal{S} \tag{17d}$$

$$w_h(s,a) \geq 0, \quad \forall(s,a,h) \in \mathcal{S} \times \mathcal{A} \times [H] \tag{17e}$$

From the optimal solution $w^*$ of (17), the optimal policy $\pi^*$ for the CMDP problem (2) can be computed using (13).

# B Extended Linear Programming Method for Solving (7) and (10)

The OFU problem (7) and the DOP problem (10) may appear much more challenging than the CMDP problem (2) because they involve a minimization over all models in $\mathcal{P}_k$, which is non-trivial. However, finding the optimistic model (and the corresponding optimistic policy) from a given confidence set is a standard step in OFU style algorithms for exploration in RL [21, 18]. In the case of standard

(unconstrained) MDP, this problem is solved using a approach called *extended value iteration* [21]. In the case of constrained MDP, (7) (and similarly (10) ) can be solved by an approach called *extended linear programming*. The details are given in [18]. We give a brief description below for completeness. Note that the description below mainly focus on solving (7). Solving (10) is identical, just by replacing the constraint cost function $c_h(\cdot, \cdot)$ with pessimist constraint cost function $\bar{c}_{h,k}(\cdot, \cdot)$, $\forall h \in [H]$, and is mentioned at the end of this subsection.

Define the state-action-state occupancy measure $z^\pi$ as $z_h^\pi(s, a, s'; P) = P_h(s'|s, a)w_h^\pi(s, a; P)$. The extended LP formulation corresponding to (7) is then given as follows:

$$\max_z \quad \sum_{s,a,s',h} z_h(s, a, s')r_h(s, a) \tag{18a}$$

$$\text{s.t.} \quad \sum_{s,a,s',h} z_h(s, a, s')c_h(s, a) \leq \bar{C} \tag{18b}$$

$$\sum_{a,s'} z_h(s, a, s') = \sum_{s',a'} z_{h-1}(s', a', s) \quad \forall h \in [H] \setminus \{1\}, s \in \mathcal{S} \tag{18c}$$

$$\sum_{a,s'} z_1(s, a, s') = \mathbb{1}\{s = s_1\}, \quad \forall s \in \mathcal{S} \tag{18d}$$

$$z_h(s, a, s') \geq 0, \quad \forall(s, a, s', h) \in \mathcal{S} \times \mathcal{A} \times \mathcal{S} \times [H], \tag{18e}$$

$$z_h(s, a, s') - (\widehat{P}_{h,k}(s'|s, a) + \beta_{h,k}^p(s, a, s')) \sum_y z_h(s, a, y) \leq 0,$$
$$\forall(s, a, s', h) \in \mathcal{S} \times \mathcal{A} \times \mathcal{S} \times [H] \tag{18f}$$

$$- z_h(s, a, s') + (\widehat{P}_{h,k}(s'|s, a) - \beta_{h,k}^p(s, a, s')) \sum_y z_h(s, a, y) \leq 0,$$
$$\forall(s, a, s', h) \in \mathcal{S} \times \mathcal{A} \times \mathcal{S} \times [H] \tag{18g}$$

The last two conditions ((18f) and (18g)) distinguish the extended LP formulation from the LP formulation for CMDP. These constraints are based on the Bernstein confidence sets around the empirical model $\widehat{P}_k$.

From the solutions $\tilde{z}^*$ of the extended LP, we can obtain the solution of (7) as

$$\underline{P}_{h,k}(s'|s, a) = \frac{\tilde{z}_h^*(s, a, s')}{\sum_y \tilde{z}_h^*(s, a, y)}, \quad \underline{\pi}_{h,k}(s, a) = \frac{\sum_{s'} \tilde{z}_h^*(s, a, s')}{\sum_{b,s'} \tilde{z}_h^*(s, b, s')}. \tag{19}$$

## C  Useful Technical Results

Here we reproduce the supporting technical results that are required for analyzing our DOPE algorithm. We begin by stating the following concentration inequality, known as empirical Bernstein inequality [30, Theorem 4].

**Lemma 7** (Empirical Bernstein Inequality). *Let $Z = (Z_1, \ldots Z_n)$ be i.i.d random vector with values in $[0, 1]^n$, and let $\delta \in (0, 1)$. Then, with probability at least $1 - \delta$, it holds that*

$$\mathbb{E}[Z] - \frac{1}{n} \sum_{i=1}^n Z_i \leq \sqrt{\frac{2V_n(Z) \log(\frac{2}{\delta})}{n}} + \frac{7 \log(\frac{2}{\delta})}{3(n-1)},$$

*where $V_n(Z)$ is the sample variance.*

We can get the following result using empirical Bernstein inequality and union bound. This result is widely used in the literature now, for example see [22, Proof of Lemma 2],

**Lemma 8.** *With probability at least $1 - 2\delta$, for all $(h, s, a, s') \in [H] \times \mathcal{S} \times \mathcal{A} \times \mathcal{S}$, $k \in [K]$, we have*

$$|P_h(s'|s, a) - \widehat{P}_{h,k}(s'|s, a)| \leq \sqrt{\frac{4\text{Var}(\widehat{P}_{h,k}(s'|s, a)) \log\left(\frac{2SAKH}{\delta}\right)}{n_{h,k}(s, a) \vee 1}} + \frac{14 \log\left(\frac{2SAKH}{\delta}\right)}{3((n_{h,k}(s, a) - 1) \vee 1)}.$$

Recall (from (4) - (5)) that

$$\beta_{h,k}^p(s,a,s') = \sqrt{\frac{4\mathrm{Var}(\widehat{P}_{h,k}(s'|s,a))\log\left(\frac{2SAKH}{\delta}\right)}{n_{h,k}(s,a)\vee 1}} + \frac{14\log\left(\frac{2SAKH}{\delta}\right)}{3(n_{h,k}(s,a)\vee 1))},$$

$$\mathcal{P}_{h,k}(s,a) = \{P' : |P_h'(s'|s,a) - \widehat{P}_{h,k}(s'|s,a)| \le \beta_{h,k}^p(s,a,s'), \forall s' \in \mathcal{S}\},$$

and define $\mathcal{P}_k = \bigcap_{(h,s,a)\in[H]\times\mathcal{S}\times\mathcal{A}} \mathcal{P}_{h,k}(s,a)$.

Define the event

$$F^p = \{P \in \mathcal{P}_k, \ \forall k \in [K]\}. \tag{20}$$

Then, using Lemma 8, we can get the following result immediately.

**Lemma 9.** *Let $F^p$ be the event defined as in (20). Then, $\mathbb{P}(F^p) \ge 1 - 2\delta$.*

Define the events $F_k^c = \{\forall(h,s,a) : |\hat{c}_{h,k}(s,a) - c_h(s,a)| \le \beta_{h,k}^l(s,a)\}$, and $F_k^r = \{\forall h,s,a : |\hat{r}_{h,k}(s,a) - r_h(s,a)| \le \beta_{h,k}^l(s,a)\}$, and define

$$F^l = \bigcap_k F_k^c \cap F_k^r \tag{21}$$

The following is a standard result, and can be obtained by Hoeffding's inequality, and using a union bound argument on all $h, s, a$ and all possible values of $n_{h,k}(s,a)$, for all $k \in [K]$.

**Lemma 10.** $\mathbb{P}(F^l) \ge 1 - 2\delta$.

We now define the event $F_w$ as follows
$F^w$

$$= \left\{ n_{h,k}(s,a) \ge \frac{1}{2}\sum_{j<k} w_{h,j}(s,a) - H\log\frac{SAH}{\delta}, \ \forall(h,s,a,s',k) \in [H]\times\mathcal{S}\times\mathcal{A}\times\mathcal{S}\times[K] \right\}, \tag{22}$$

where $w_{h,j}$ is the occupancy measure corresponding to the policy chosen in episode $j$. We have the following result from [14, Corollary E.4.]

**Lemma 11** (Corollary E.4., [14]). *Let $F^w$ be the event defined as in (22). Then, $\mathbb{P}(F^w) \ge 1 - \delta$.*

We now define the **good event** $G = F^p \cap F^w \cap F^l$. Using union bound, we can show that $\mathbb{P}(G) \ge 1 - 5\delta$. Since our analysis is based on this good event, we formally state it as a lemma.

**Lemma 12.** *Let $F^p$ is defined as in (20) and $F^l$ defined in (21), $F^w$ is defined as in (22). Let the good event $G = F^p \cap F^w \cap F^l$. Then, $\mathbb{P}(G) \ge 1 - 5\delta$.*

We will also use the following results for analyzing the performance of our DOPE algorithm.

**Lemma 13** (Lemma 36, [18]). *Under the event $F^w$,*

$$\sum_{k=1}^K \sum_{h=1}^H \mathbb{E}\left[ \frac{1}{\sqrt{n_{h,k}(s_{h,k},a_{h,k})\vee 1}} \Big| \mathcal{F}_{k-1} \right] \le \tilde{\mathcal{O}}(\sqrt{SAH^2K} + SAH).$$

**Lemma 14** (Lemma 37, [18]). *Under the event $F^w$,*

$$\sum_{k=1}^K \sum_{h=1}^H \mathbb{E}\left[ \frac{1}{n_{h,k}(s_{h,k},a_{h,k})\vee 1} \Big| \mathcal{F}_{k-1} \right] \le \tilde{\mathcal{O}}(SAH^2).$$

**Lemma 15** (Lemma 8,[22]). *Under the event $G$, for all $k, h, s, a, s'$, and for all $P' \in \mathcal{P}_k$, there exists constants $C_1, C_2 > 0$ such that $|P_h'(s'|s,a) - P_h(s'|s,a)| \le C_1\sqrt{\frac{P_h(s'|s,a)L}{n_{h,k}(s,a)\vee 1}} + C_2\frac{L}{n_{h,k}(s,a)\vee 1}$.*

**Lemma 16** (Value difference lemma). *Consider two MDPs* $M = (\mathcal{S}, \mathcal{A}, l, P)$ *and* $M' = (\mathcal{S}, \mathcal{A}, l', P')$. *For any policy* $\pi$, *state* $s \in \mathcal{S}$, *and time step* $h \in [H]$, *the following relation holds.*

$$V_{l,h}^{\pi}(s; P) - V_{l',h}^{\pi}(s; P')$$

$$= \mathbb{E}\left[\sum_{\tau=h}^{H}(l_{\tau}(s_{\tau}, a_{\tau}) - l'_{\tau}(s_{\tau}, a_{\tau})) + ((P_{\tau} - P'_{\tau})(\cdot|s_{\tau}, a_{\tau}))^{\top}V_{l,\tau+1}^{\pi}(\cdot; P)|s_h = s, \pi, P'\right]$$

$$= \mathbb{E}\left[\sum_{\tau=h}^{H}(l'_{\tau}(s_{\tau}, a_{\tau}) - l_{\tau}(s_{\tau}, a_{\tau})) + ((P'_{\tau} - P_{\tau})(\cdot|s_{\tau}, a_{\tau}))^{\top}V_{l,\tau+1}^{\pi}(\cdot; P')|s_h = s, \pi, P\right].$$

## D  Proof of the Main Results

All the results we prove in this section are conditioned on the good event $G$ defined in Section C. So, the results hold true with a probability greater than $1 - 5\delta$ according to Lemma 12. We will omit stating this conditioning under $G$ in each statement to avoid repetition.

### D.1  Proofs of Proposition 4

First note that

$$\mathbb{E}[\sum_{h=1}^{H}\bar{c}_{h,k}(s_h, a_h)|\pi', P', \mathcal{F}_{k-1}] = \mathbb{E}[\sum_{h=1}^{H}\left(\hat{c}_{h,k}(s_h, a_h) + \beta_{h,k}^{l}(s_h, a_h) + H\bar{\beta}_{h,k}^{p}(s_h, a_h)\right)|\pi', P']$$

$$= V_{\hat{c}_k}^{\pi'}(P') + \eta_k^{\pi'}(P') + \epsilon_k^{\pi'}(P'), \tag{23}$$

$$\mathbb{E}[\sum_{h=1}^{H}\bar{r}_{h,k}(s_h, a_h)|\pi', P', \mathcal{F}_{k-1}] = \mathbb{E}[\sum_{h=1}^{H}\left(\hat{r}_{h,k}(s_h, a_h) + \frac{3H}{\bar{C} - \bar{C}_b}\beta_{h,k}^{l}(s^h, a^h)\right.$$

$$\left. - \frac{H^2}{\bar{C} - \bar{C}_b}\bar{\beta}_{h,k}^{p}(s_h, a_h)\right)|\pi', P']$$

$$= V_{\hat{r}_k}^{\pi'}(P') - \frac{3H}{\bar{C} - \bar{C}_b}\eta_k^{\pi'}(P') - \frac{H}{\bar{C} - \bar{C}_b}\epsilon_k^{\pi'}(P'), \tag{24}$$

where equations (23), (24) are due to linearity of expectation.

**Lemma 17.** *Let* $\epsilon_k^{\pi'}(P')$ *and* $\eta_k^{\pi'}(P')$ *be as defined in* (11). *Also, let* $\{\pi_k\}$ *be the sequence of policies generated by DOPE algorithm. Then, for any* $K' \leq K$, *each of the following relations hold with a probability greater than* $1 - 5\delta$.

$$\sum_{k=1}^{K'}\epsilon_k^{\pi_k}(P) \leq \tilde{\mathcal{O}}(S\sqrt{AH^4K'}), \text{ and, } \sum_{k=1}^{K'}\eta_k^{\pi_k}(P) \leq \tilde{\mathcal{O}}(S\sqrt{AH^2K'}).$$

*Proof.*

$$\sum_{k=1}^{K'}\epsilon_k^{\pi_k}(P) = H\sum_{k=1}^{K'}\sum_{h=1}^{H}\mathbb{E}[\sum_{s'}\beta_{h,k}^{p}(s_{h,k}, a_{h,k}, s')|\pi_k, P, \mathcal{F}_{k-1}]$$

$$\overset{(a)}{\leq} H\sum_{k=1}^{K'}\mathbb{E}\left[\sum_{h=1}^{H}\sqrt{\frac{4L}{n_{h,k}(s_{h,k}, a_{h,k}) \vee 1}}\sum_{s' \in \mathcal{S}}\sqrt{\widehat{P}_{h,k}(s'|s_{h,k}, a_{h,k})}|\pi_k, P, \mathcal{F}_{k-1}\right]$$

$$+ HS\sum_{k=1}^{K'}\mathbb{E}\left[\sum_{h=1}^{H}\frac{(14/3)L}{n_{h,k}(s_{h,k}, a_{h,k}) \vee 1}|\pi_k, P, \mathcal{F}_{k-1}\right]$$

$$\overset{(b)}{\leq} 2H\sqrt{S}\sqrt{L}\sum_{k=1}^{K'}\mathbb{E}\left[\sum_{h=1}^{H}\sqrt{\frac{1}{n_{h,k}(s_{h,k}, a_{h,k}) \vee 1}}|\pi_k, P, \mathcal{F}_{k-1}\right]$$

$$+ (14/3)HSL\sum_{k=1}^{K'}\mathbb{E}\left[\sum_{h=1}^{H}\frac{1}{n_{h,k}(s_{h,k}, a_{h,k}) \vee 1}|\pi_k, P, \mathcal{F}_{k-1}\right]$$

$$\overset{(c)}{\leq} H\sqrt{S}\sqrt{L}\tilde{\mathcal{O}}(\sqrt{SAH^2K'} + SAH) + HSL\tilde{\mathcal{O}}(SAH) \leq \tilde{\mathcal{O}}(S\sqrt{AH^4K'}). \tag{25}$$

Here, we get inequality $(a)$ by the definition of $\beta_{h,k}^p$ (c.f. (5)). To get $(b)$, note that $\sum_{s' \in \mathcal{S}} \sqrt{\widehat{P}_{h,k}(s'|(s_{h,k}, a_{h,k}))} \leq \sqrt{\sum_{s'} \widehat{P}_{h,k}(s'|(s_{h,k}, a_{h,k}))}\sqrt{S}$ by Cauchy-Schwarz inequality and $\sum_{s'} \widehat{P}_{h,k}(s'|(s_{h,k}, a_{h,k})) = 1$. We get $(c)$ using Lemma 13 and Lemma 14.

The other part can also be obtained similarly from Lemma 13. □

We now give the proof of Proposition 4.

***Proof of Proposition 4.*** First note that even though $(\pi_b, P)$ is a feasible solution for the original CMDP problem (2), it may not feasible for the DOP problem (10). To see this, note that since $V_{\bar{c}_k}^{\pi_b}(P) = V_{\hat{c}_k}^{\pi_b}(P) + \eta_k^{\pi_b}(P) + \epsilon_k^{\pi_b}(P)$ and $V_{\hat{c}_k}^{\pi_b}(P) \leq V_c^{\pi_b}(P) + \eta_k^{\pi_b}(P)$, and $V_c^{\pi_b}(P) = \bar{C}_b$, we will have $V_{\bar{c}_k}^{\pi_b}(P) \leq \bar{C}$ if $2\eta_k^{\pi_b}(P) + \epsilon_k^{\pi_b}(P) \leq (C - \bar{C}_b)$. So, $(\pi_b, P)$ is a feasible solution for (10) if, $2\eta_k^{\pi_b}(P) + \epsilon_k^{\pi_b}(P) \leq (C - \bar{C}_b)$. This is a sufficient condition for the feasibility of $(\pi_b, P)$. This condition may not be satisfied in the initial episodes.

However, since $\eta_k^{\pi_b}(P)$ and $\epsilon_k^{\pi_b}(P)$ are decreasing in $k$, if $(\pi_b, P)$ becomes a feasible solution for (10) at episode $k'$, then it will remain to be a feasible solution for all episodes $k \geq k'$.

Suppose $\pi_k = \pi_b$ for all $k \leq K'$. Also, suppose the above condition is not satisfied in the algorithm until episode $K' + 1$. Then, $2\eta_k^{\pi_b}(P) + \epsilon_k^{\pi_b}(P) > C - \bar{C}_b$ for all $k \leq K'$. So, we should get

$$K'(\bar{C} - \bar{C}_b) < \sum_{k=1}^{K'} 2\eta_k^{\pi_b}(P) + \epsilon_k^{\pi_b}(P) = \sum_{k=1}^{K'} 2\eta_k^{\pi_k}(P) + \epsilon_k^{\pi_k}(P) \leq \tilde{\mathcal{O}}(S\sqrt{AH^4 K'}),$$

where the last inequality is from Lemma 17. However, this inequality is violated for $K' \geq \tilde{\mathcal{O}}(\frac{S^2 AH^4}{(\bar{C} - \bar{C}_b)^2})$. So, $(\pi_b, P)$ is a feasible solution for (10) for any episode $k \geq K_o = \tilde{\mathcal{O}}(\frac{S^2 AH^4}{(\bar{C} - \bar{C}_b)^2})$ provided that $\pi_k = \pi_b$ for all $k \leq K_o$. □

The above result, however, only shows that $\pi_b$ becomes a feasible policy after some finite number of episodes. A natural question is, is $\pi_b$ the only feasible policy? In such a case, the DOPE algorithm may not provide enough exploration to learn the optimal policy.

We alleviate the concerns about the above possible issue by showing that for all $k \geq K_o$, there exists a feasible solution $(\pi', P)$ for the OP problem (10) such that $w_h^{\pi'}(s, a; P) > 0$ for every $(s, a) \in \mathcal{S} \times \mathcal{A}$ with $w_h^{\pi^*}(s, a; P) > 0$. Informally, this implies that $\pi'$ will visit all state-action pairs that will be visited by the optimal policy $\pi^*$. This result can be derived as a corollary for Proposition 4.

## D.2 Proof of Proposition 5

*Proof.* For any episode $k \leq K_o$, we have $\pi_k = \pi_b$, and it is safe by Assumption 1. For $k \geq K_o$, (10) is feasible according to Proposition 4. Since $(\pi_k, P_k)$ is the solution of (10), we have $V_{\bar{c}_k}^{\pi_k}(P_k) \leq C$. We will now show that $V_c^{\pi_k}(P) \leq C$, conditioned on the good event $G$.

By the value difference lemma (Lemma 16), we have

$$V_c^{\pi_k}(P) - V_c^{\pi_k}(P_k) = \mathbb{E}[\sum_{h=1}^{H} ((P_h - P_{h,k})(\cdot|s_{h,k}, a_{h,k}))^\top V_{c,h+1}^{\pi_k}(\cdot; P) | \pi_k, P_k, \mathcal{F}_{k-1}]$$

$$\overset{(a)}{\leq} \mathbb{E}[\sum_{h=1}^{H} \|((P_h - P_{h,k})(\cdot|s_{h,k}, a_{h,k}))\|_1 \|V_{c,h+1}^{\pi_k}(\cdot; P)\|_\infty | \pi_k, P_k, \mathcal{F}_{k-1}]$$

$$\overset{(b)}{\leq} H\mathbb{E}[\sum_{h=1}^{H} \bar{\beta}_{h,k}^p(s_{h,k}, a_{h,k}) | \pi_k, P_k, \mathcal{F}_{k-1}] = \epsilon_k^{\pi_k}(P_k). \tag{26}$$

Here, we get $(a)$ by Holder's inequality inequality. To get $(b)$, we make use of two observations. First, note that $\|V_{c,h+1}^{\pi_k}(\cdot; P)\|_\infty \leq H$ because the expected cumulative cost cannot be grater than $H$ since $|c(\cdot, \cdot)| \leq 1$ by assumption. Second, under the good event $G$, $\sum_{s'} |P_h(s'|s, a) - P_{h,k}(s'|s, a)| \leq \sum_{s'} \beta_{h,k}^p(s, a, s') = \bar{\beta}_{h,k}^p(s, a)$.

From (26), we get

$$V_c^{\pi_k}(P) \le V_c^{\pi_k}(P_k) + \epsilon_k^{\pi_k}(P_k) \overset{(c)}{\le} V_{\hat{c}_k}^{\pi_k}(P_k) + \eta_k^{\pi_k}(P_k) + \epsilon_k^{\pi_k}(P_k) = V_{\bar{c}_k}^{\pi_k}(P_k) \overset{(d)}{\le} \bar{C},$$

where $(c)$ is by definition of good event and $(d)$ is from the fact that $(\pi_k, P_k)$ is the solution of (10). So, $V_c^{\pi_k}(P) \le \bar{C}$, and hence $\pi_k$ is safe, under the good event $G$. So, this statement holds with a probability greater than $1 - 5\delta$, according to Lemma 12. $\qquad\square$

### D.3 Proof of Theorem 3

We first prove an important lemma.

**Lemma 18** (Optimism). *Let $(\pi_k, P_k)$ be the optimal solution corresponding to the DOP problem (10). Then,*

$$V_{\tilde{r}_k}^{\pi_k}(P_k) \le V_r^{\pi^*}(P).$$

*Proof.* We will first consider a more general version of the DOP problem (10) as

$$(\tilde{\pi}_k, \tilde{P}_k) = \underset{\pi', P' \in \mathcal{P}_k}{\arg\min} \; V_{\tilde{r}_k}^{\pi'}(P') \text{ subject to } V_{\bar{c}_k}^{\pi'}(P') \le \bar{C}, \tag{27}$$

where we change $\bar{r}_k$ in (10) to $\tilde{r}_k$ above, with $\tilde{r}_h^k(s,a) = \hat{r}_h(s,a) - 3b\beta_{h,k}^l(s,a) - bH\bar{\beta}_{h,k}^p(s,a)$, for $b > 0$. Note that (27) reduces to (10) for $b = \frac{H}{\bar{C} - \bar{C}_b}$ and hence it is indeed a general version.

Using the occupancy measures $w_h^{\pi_b}$ and $w_h^{\pi^*}$, define a new occupancy measure $\tilde{w}_h(s,a) = (1 - \alpha_k)w_h^{\pi_b}(s,a;P) + \alpha_k w_h^{\pi^*}(s,a;P)$ for an $\alpha_k > 0$.

Note that $\tilde{w}$ is a valid occupancy measure since the set of occupancy measure is convex (c.f. Proposition 6). Let $\tilde{\pi}$ be the policy corresponding to the occupancy measure $\tilde{w}$, which can be obtained according to (13) so that $\tilde{w} = w^{\tilde{\pi}}$.

**Claim 1:** $(\tilde{\pi}, P)$ is a feasible solution for (27) when $\alpha_k$ satisfies the sufficient condition

$$\alpha_k \le \frac{\bar{C} - \bar{C}_b - (\epsilon_k^{\pi_b}(P) + 2\eta_k^{\pi_b}(P))}{\bar{C} - \bar{C}_b + (\epsilon_k^{\pi^*}(P) + 2\eta_k^{\pi^*}(P)) - (\epsilon_k^{\pi_b}(P) + 2\eta_k^{\pi_b}(P))}. \tag{28}$$

*Proof of Claim 1:* Since value function is a linear function of the occupancy measure, we have

$$V_{\bar{c}_k}^{\tilde{\pi}}(P) = (1 - \alpha_k)V_{\bar{c}_k}^{\pi_b}(P) + \alpha_k V_{\bar{c}_k}^{\pi^*}(P)$$

$$= (1 - \alpha_k)(V_{\hat{c}_k}^{\pi_b}(P) + \eta_k^{\pi_b}(P) + \epsilon_k^{\pi_b}(P)) + \alpha_k(V_{\hat{c}_k}^{\pi^*}(P) + \eta_k^{\pi^*}(P) + \epsilon_k^{\pi^*}(P))$$

$$\overset{(a)}{\le} (1 - \alpha_k)(V_c^{\pi_b}(P) + 2\eta_k^{\pi_b}(P) + \epsilon_k^{\pi_b}(P)) + \alpha_k(V_c^{\pi^*}(P) + 2\eta_k^{\pi^*}(P) + \epsilon_k^{\pi^*}(P))$$

$$\overset{(b)}{\le} (1 - \alpha_k)(\bar{C}_b + 2\eta_k^{\pi_b}(P) + \epsilon_k^{\pi_b}(P)) + \alpha_k(\bar{C} + 2\eta_k^{\pi^*}(P) + \epsilon_k^{\pi^*}(P)),$$

where inequality $(a)$ is due to the good event that $c$ is within the confidence interval, $V_{\hat{c}_k}^{\pi}(P) - \eta_k^{\pi}(P) \le V_c^{\pi}(P)$, for any $\pi$ and inequality $(b)$ is due to the fact that $V_c^{\pi_b}(P) = \bar{C}_b$, and $V_c^{\pi^*}(P) \le \bar{C}$.

For $(\tilde{\pi}_k, P)$ to be a feasible solution for (27), it must be true that $V_{\bar{c}_k}^{\tilde{\pi}}(P) \le \bar{C}$. Hence, it is sufficient to get an $\alpha_k$ such that

$$(1 - \alpha_k)(\bar{C}_b + 2\eta_k^{\pi_b}(P) + \epsilon_k^{\pi_b}(P)) + \alpha_k(\bar{C} + 2\eta_k^{\pi^*}(P) + \epsilon_k^{\pi^*}(P)) \le \bar{C}.$$

This yields a sufficient condition (28). Note that $\alpha_k$ is non-negative because $2\eta_k^{\pi_b}(P) + \epsilon_k^{\pi_b}(P) \le \bar{C} - \bar{C}_b$ for $k \ge K_o$, as shown in the proof of Proposition 4. This concludes the proof of Claim 1.

**Claim 2:** $V_{\tilde{r}_k}^{\tilde{\pi}_k}(\tilde{P}_k) \le V_r^{\pi^*}(P)$ if $b$ satisfies the sufficient condition

$$b \ge \frac{H}{\bar{C} - \bar{C}_b}. \tag{29}$$

*Proof of Claim 2:* Selecting an $\alpha_k$ that satisfies the condition (28), $(\tilde{\pi}, P)$ is a feasible solution of (27). Since $(\tilde{\pi}_k, \tilde{P}_k)$ is the optimal solution of (27), we have $V_{\tilde{r}_k}^{\tilde{\pi}_k}(\tilde{P}_k) \le V_{\tilde{r}_k}^{\tilde{\pi}}(P)$. So, it is sufficient to find a $b$ such that $V_{\tilde{r}_k}^{\tilde{\pi}}(P) \le V_r^{\pi^*}(P)$. Using the linearity of the value function w.r.t. occupancy measure, this is equivalent to $(1-\alpha_k)(V_{\hat{r}_k}^{\pi_b}(P) - 3b\eta_k^{\pi_b}(P) - b\epsilon_k^{\pi_b}(P)) + \alpha_k(V_{\hat{r}_k}^{\pi^*}(P) - 3b\eta_k^{\pi^*}(P) - b\epsilon_k^{\pi^*}(P)) \le V_r^{\pi^*}(P).$

Since $V_{\hat{r}_k}^{\pi}(P) - b\eta_k^{\pi}(P) \le V_{\hat{r}_k}^{\pi}(P) - \eta_k^{\pi}(P) \le V_r^{\pi}(P)$ for any $\pi$ under the good event, it is sufficient if we find a $b$ such that $(1-\alpha_k)(V_r^{\pi_b}(P) - 2b\eta_k^{\pi_b}(P) - b\epsilon_k^{\pi_b}(P)) + \alpha_k(V_r^{\pi^*} - 2b\eta_k^{\pi^*}(P) - b\epsilon_k^{\pi^*}(P)) \le$

$V_r^{\pi^*}(P)$. This will yield the condition $b \geq \frac{V_r^{\pi^b}(P) - V_r^{\pi^*}(P)}{[\epsilon_k^{\pi^b}(P) + 2\eta_k^{\pi^b}(P)] + \frac{\alpha_k}{1-\alpha_k}[\epsilon_k^{\pi^*}(P) + 2\eta_k^{\pi^*}(P)]}$. Now, we choose $\alpha_k$ that satisfies the condition (28) as, $\frac{\alpha_k}{1-\alpha_k} = \frac{\bar{C} - \bar{C}_b - [\epsilon_k^{\pi^b}(P) + 2\eta_k^{\pi^b}(P)]}{\epsilon_k^{\pi^*}(P) + 2\eta_k^{\pi^b}(P)}$. Using this in the previous inequality for $b$, we get the sufficient condition $b \geq \frac{V_r^{\pi^b}(P) - V_r^{\pi^*}(P)}{\bar{C} - \bar{C}_b}$. Since $V_r^{\pi^b}(P) \leq H$ and $V_r^{\pi^*}(P) \geq 0$, we get the sufficient condition (29). This concludes the proof of Claim 2.

Now, let $b = \frac{H}{\bar{C} - \bar{C}_b}$. So, $\tilde{r}_k = \bar{r}_k$ and $(\tilde{\pi}_k, \tilde{P}_k) = (\pi_k, P_k)$. Hence, by Claim 2, we have $V_{\bar{r}_k}^{\pi_k}(P_k) \leq V_r^{\pi^*}(P)$. Hence, we have the desired result. $\qquad\square$

We now present the proof of Theorem 3.

***Proof of Theorem 3.*** The regret for the DOPE algorithm after $K$ episodes can be written as,
$$R(K) = \sum_{k=1}^{K_o}(V_r^{\pi_k}(P) - V_r^{\pi^*}(P)) + \sum_{k=K_o}^{K}(V_r^{\pi_k}(P) - V_r^{\pi^*}(P)). \tag{30}$$
We will bound the first term in (30) as
$$\sum_{k=1}^{K_o}(V_r^{\pi_k}(P) - V_r^{\pi^*}(P)) \leq HK_0 \leq \tilde{\mathcal{O}}\left(\frac{S^2 A H^5}{(\bar{C} - \bar{C}_b)^2}\right), \tag{31}$$
where we get the first inequality because $(V_r^{\pi_k}(P) - V_r^{\pi^*}(P)) \leq H$, and the second inequality follows from the bound on $K_o$ in Proposition 4.

The second term in (30) can be bounded as
$$\sum_{k=K_o}^{K}(V_r^{\pi_k}(P) - V_r^{\pi^*}(P)) = \sum_{k=K_o}^{K}(V_r^{\pi_k}(P) - V_{\bar{r}_k}^{\pi_k}(P_k)) + \sum_{k=K_o}^{K}(V_{\bar{r}_k}^{\pi_k}(P_k) - V_r^{\pi^*}(P))$$

$$\overset{(a)}{\leq} \sum_{k=K_o}^{K}(V_r^{\pi_k}(P) - V_{\bar{r}_k}^{\pi_k}(P_k)) = \sum_{k=K_o}^{K}(V_r^{\pi_k}(P) - V_{\hat{r}_k}^{\pi_k}(P)) + \sum_{k=K_o}^{K}(V_{\hat{r}_k}^{\pi_k}(P) - V_{\bar{r}_k}^{\pi_k}(P_k))$$

$$\overset{(b)}{\leq} \sum_{k=K_o}^{K}\eta_k^{\pi_k}(P) + \sum_{k=K_o}^{K}(V_{\hat{r}_k}^{\pi_k}(P) - V_{\bar{r}_k}^{\pi_k}(P_k))$$

$$\overset{(c)}{\leq} \sum_{k=K_o}^{K}\eta_k^{\pi_k}(P) + \sum_{k=K_0}^{K}(V_{\bar{r}_k}^{\pi_k}(P) - V_{\bar{r}_k}^{\pi_k}(P_k)) + \frac{H}{\bar{C} - \bar{C}_b}\sum_{k=K_0}^{K}(3\eta_k^{\pi_k}(P) + \epsilon_k^{\pi_k}(P))$$

$$\overset{(d)}{\leq} \sum_{k=K_o}^{K}(V_{\bar{r}_k}^{\pi_k}(P) - V_{\bar{r}_k}^{\pi_k}(P_k)) + \tilde{O}\left(\frac{H^3}{\bar{C} - \bar{C}_b}S\sqrt{AK}\right), \tag{32}$$
where $(a)$ is due to the fact that $V_{\bar{r}_k}^{\pi_k}(P_k) \leq V_r^{\pi^*}(P)$ from Lemma 18, $(b)$ is due to the fact that $|r_h(s,a) - \hat{r}_{h,k}(s,a)| \leq \beta_{h,k}^l(s,a)$ conditioned on the good event set $G$ (see Lemma 10), $(c)$ follows from the definition of $\bar{r}_k$, and $(d)$ follows from Lemma 17.

We will now bound the first term in (32) as
$$\sum_{k=K_0}^{K}(V_{\bar{r}_k}^{\pi_k}(P) - V_{\bar{r}_k}^{\pi_k}(P_k))$$

$$\overset{(e)}{=} \sum_{k=K_0}^{K}\sum_{h=1}^{H}\mathbb{E}\left[\sum_{s'}(P_h - P_{h,k})(s'|s_{h,k}, a_{h,k})V_{\bar{r}_k, h+1}^{\pi_k}(s'; P_k)|\pi_k, P, \mathcal{F}_{k-1}\right]$$

$$\leq \sum_{k=K_0}^{K}\sum_{h=1}^{H}\mathbb{E}\left[\sum_{s'}|(P_h - P_{h,k})(s'|s_{h,k}, a_{h,k})||V_{r, h+1}^{\pi_k}(s'; P)||\pi_k, P, \mathcal{F}_{k-1}\right] +$$

$$\sum_{k=K_0}^{K}\sum_{h=1}^{H}\mathbb{E}\left[|\sum_{s'}(P_h - P_{h,k})(s'|s_{h,k}, a_{h,k})(V_{\bar{r}_k, h+1}^{\pi_k}(s'; P_k) - V_{r, h+1}^{\pi_k}(s'; P))||\pi_k, P, \mathcal{F}_{k-1}\right] \tag{33}$$

Now, for the first term in (33),

$$\sum_{k=K_0}^{K}\sum_{h=1}^{H}\mathbb{E}[\sum_{s'}|(P_h - P_{h,k})(s'|s_{h,k}, a_{h,k})||V_{r,h+1}^{\pi_k}(s'; P)|| \pi_k, P, \mathcal{F}_{k-1}]$$

$$\leq \sum_{k=K_0}^{K}\epsilon_k^{\pi_k}(P) \overset{(f)}{\leq} \tilde{O}(S\sqrt{AH^4K'}), \qquad (34)$$

where $(e)$ is obtained from the value difference lemma (Lemma 16), and $(f)$ is from Lemma 17.

In order to now bound the second term in (33), we proceed in similar lines to the proof of Lemma 32 from [18].

Consider,

$$\sum_{k=K_0}^{K}\sum_{h=1}^{H}\mathbb{E}[|\sum_{s'}(P_h - P_{h,k})(s'|s_{h,k}, a_{h,k})(V_{\tilde{r}_k,h+1}^{\pi_k}(s'; P_k) - V_{r,h+1}^{\pi_k}(s'; P))|| \pi_k, P, \mathcal{F}_{k-1}]$$

$$= \sum_{k,h,s,a} w_h^{\pi_k}(s, a; P) \sum_{s'}(P_h - P_{h,k})(s'|s_{h,k}, a_{h,k})(V_{\tilde{r}_k,h+1}^{\pi_k}(s'; P_k) - V_{r,h+1}^{\pi_k}(s'; P))$$

$$\leq \underbrace{\sum_{k,h,s,a} w_h^{\pi_k}(s, a; P) \sum_{s'}\frac{\sqrt{P_h(s'|s, a)}}{\sqrt{n_{h,k}(s, a)\vee 1}}|V_{\tilde{r}_k,h+1}^{\pi_k}(s'; P_k) - V_{r,h+1}^{\pi_k}(s'; P)|}_{(A)}$$

$$+ \underbrace{\sum_{k,h,s,a} w_h^{\pi_k}(s, a; P)\frac{1}{n_{h,k}(s, a)\vee 1}|V_{\tilde{r}_k,h+1}^{\pi_k}(s'; P_k) - V_{r,h+1}^{\pi_k}(s'; P)|}_{(B)}, \qquad (35)$$

where the last inequality is obtained from Lemma 15. We will bound the term $(B)$ in (35) as

$$(B) \leq \frac{H^3 SL}{\bar{C} - \bar{C}_b}\sum_{k,h,s,a} w_h^{\pi_k}(s, a; P)\frac{1}{n_{h,k}(s, a)\vee 1} \leq \frac{H^5 S^2 AL}{\bar{C} - \bar{C}_b}, \qquad (36)$$

where the first inequality is from bounding $|V_{\tilde{r}_k}^{\pi_k}(P_k) - V_r^{\pi_k}(P)|$ by $\frac{H^3 SL}{\bar{C} - \bar{C}_b}$. This is obtained by noting that $V_{\tilde{r}_k}^{\pi_k}(P_k) = V_{\hat{r}_k}^{\pi_k}(P_k) - \frac{H}{\bar{C} - \bar{C}_b}\eta_k^{\pi_k}(P_k) - \frac{H^2}{\bar{C} - \bar{C}_b}\epsilon_k^{\pi_k}(P_k) \leq V_{\hat{r}_k}^{\pi_k}(P_k) + \frac{H}{\bar{C} - \bar{C}_b}\mathbb{E}[\sum_{h=1}^{H}\beta_{h,k}^l(s_{h,k}, a_{h,k})|\pi_k, P_k] + \frac{H^2}{\bar{C} - \bar{C}_b}\mathbb{E}[\sum_{h=1}^{H}\sum_{s'}\beta_{h,k}^p(s_{h,k}, a_{h,k}, s')|\pi_k, P_k] \leq \frac{H^3 SL}{\bar{C} - \bar{C}_b}$, since $\sum_{h=1}^{H}\sum_{s'}\beta_{h,k}^p(s_{h,k}, a_{h,k}, s') \leq HSL$, from the definition. The second inequality is from Lemma 17.

We now bound the term $(A)$ in (35) as follows.

$$(A) \overset{(a)}{\leq} \sum_{k,h,s,a} w_h^{\pi_k}(s, a; P)\frac{\sqrt{S\sum_{s'}P_h(s'|s, a)(V_{\tilde{r}_k,h+1}^{\pi_k}(s'; P_k) - V_{r,h+1}^{\pi_k}(s'; P))^2}}{\sqrt{n_{h,k}(s, a)\vee 1}}$$

$$\overset{(b)}{\leq}\sqrt{S}\left(\sum_{k,h,s,a}\frac{w_h^{\pi_k}(s, a; P)}{n_{h,k}(s, a)\vee 1}\right)^{1/2}\left(\sum_{k,h,s,a,s'}w_h^{\pi_k}(s, a; P)P_h(s'|s, a)(V_{\tilde{r}_k,h+1}^{\pi_k}(s'; P_k) - V_{r,h+1}^{\pi_k}(s'; P))^2\right)^{1/2}$$

$$\overset{(c)}{=}\sqrt{S}\left(\sum_{k,h,s,a}\frac{w_h^{\pi_k}(s, a; P)}{n_{h,k}(s, a)\vee 1}\right)^{1/2}\left(\sum_{k,h,s',a}w_{h+1}^{\pi_k}(s', a; P)(V_{\tilde{r}_k,h+1}^{\pi_k}(s'; P_k) - V_{r,h+1}^{\pi_k}(s'; P))^2\right)^{1/2}$$

$$\overset{(d)}{\leq}\sqrt{S}\sqrt{SAH^2}\left(\sum_{k,h,s,a}w_{h+1}^{\pi_k}(s, a; P)(V_{r,h+1}^{\pi_k}(s; P) - V_{\tilde{r}_k,h+1}^{\pi_k}(s; P_k))^2\right)^{1/2}$$

$$\overset{(e)}{\leq} \sqrt{S}\sqrt{SAH^2}\frac{H^3 SL}{\bar{C}-\bar{C}_b}\left(\sum_{k,h,s,a} w_{h+1}^{\pi_k}(s,a;P)(V_{r,h+1}^{\pi_k}(s;P)-V_{\bar{r}_k,h+1}^{\pi_k}(s;P_k))\right)^{1/2}$$

$$\overset{(f)}{\leq} \frac{S^2 H^{4.5}\sqrt{AL}}{\bar{C}-\bar{C}_b}\left(\sum_k (V_r^{\pi_k}(s_1;P)-V_{\bar{r}_k}^{\pi_k}(s_1;P_k))\right.$$
$$\left. + \sum_{k,h,s,a} w_h^{\pi_k}(s,a;P)|\langle(P_h-P_{h,k})(.|s,a),(V_{\bar{r}^k,h+1}^{\pi_k}(\cdot;P_k)-V_{r,h+1}^{\pi_k}(\cdot;P))\rangle|\right)^{1/2}$$

$$\overset{(g)}{\leq} \frac{S^2 H^{4.5}\sqrt{AL}}{\bar{C}-\bar{C}_b}\left(\sum_k (V_r^{\pi_k}(s_1;P)-V_{\bar{r}^k}^{\pi_k}(s_1;P_k))\right)^{1/2}$$
$$+ \frac{S^2 H^{4.5}\sqrt{AL}}{\bar{C}-\bar{C}_b}\left(\sum_{k,h,s,a} w_h^{\pi_k}(s,a;P)|\langle(P_h-P_{h,k})(.|s,a),(V_{\bar{r}^k,h+1}^{\pi_k}(\cdot;P_k)-V_{r,h+1}^{\pi_k}(\cdot;P))\rangle|\right)^{1/2}.$$
$$(37)$$

Here, $(a)$ is obtained by Jensen's inequality, $(b)$ is by cauchy schwartz inequality, $(c)$ is from the property of the occupancy measure, i.e., $\sum_{s,a} P_h(s'|s,a)w_h(s,a,P) = \sum_a w_{h+1}(s',a,P)$, $(d)$ is obtained from Lemma 14. To get $(e)$, we use the result from Lemma 19 that $V_{\bar{r}_k,h+1}^{\pi_k}(P_k) \leq V_{r,h+1}^{\pi_k}(P)$, and hence obtain, $(V_{\bar{r}_k,h+1}^{\pi_k}(P_k) - V_{r,h+1}^{\pi_k}(P)(s))^2 \leq \frac{H^3 SL}{\bar{C}-\bar{C}_b}(V_{r,h+1}^{\pi_k}(P) - V_{\bar{r}_k,h+1}^{\pi_k}(P_k)(s))$. We prove $(e)$ from Lemma 33 of [18]. The step is to get $(f)$ is more involved and we prove it separately in Lemma 20, following a similar result from Lemma 33 of [18]. The inequality $(g)$ holds from the fact that $\sqrt{a+b} \leq \sqrt{a} + \sqrt{b}$.

Using the above obtained bounds on $(A)$ and $(B)$ in (35), we get
$$\sum_{k,h,s,a} w_h^{\pi_k}(s,a;P)|\langle(P_h-P_{h,k})(.|s,a),(V_{\bar{r}_k,h+1}^{\pi_k}(\cdot;P_k)-V_{r,h+1}^{\pi_k}(\cdot;P))\rangle|$$

$$\leq \frac{H^5 S^2 AL}{\bar{C}-\bar{C}_b} + \frac{S^2 H^{4.5}\sqrt{AL}}{\bar{C}-\bar{C}_b}\left(\sum_k (V_r^{\pi_k}(s_1;P)-V_{\bar{r}_k}^{\pi_k}(s_1;P_k))\right)^{1/2}$$

$$+ \frac{S^2 H^{4.5}\sqrt{AL}}{\bar{C}-\bar{C}_b}\left(\sum_{k,h,s,a} w_h^{\pi_k}(s,a;P)|\langle(P_h-P_{h,k})(.|s,a)(V_{\bar{r}_k,h+1}^{\pi_k}(\cdot;P_k)-V_{r,h+1}^{\pi_k}(\cdot;P))\rangle|\right)^{1/2}.$$

Let $X = \sum_{k,h,s,a} w_h^{\pi_k}(s,a,P)|\langle(P_h-P_{h,k})(.|s,a)(V_{\bar{r}_k,h+1}^{\pi_k}(\cdot;P_k)-V_{r,h+1}^{\pi_k}(\cdot;P))\rangle|$. Then, the above bound takes the form $0 \leq X \leq a + b\sqrt{X}$, where $a = \frac{H^5 S^2 AL}{\bar{C}-\bar{C}_b} + \frac{S^2 H^{4.5}\sqrt{AL}}{\bar{C}-\bar{C}_b}\left(\sum_k (V_r^{\pi_k}(s_1;P)-V_{\bar{r}_k}^{\pi_k}(s_1;P_k))\right)^{1/2}$, and $b = \frac{S^2 H^{4.5}\sqrt{AL}}{\bar{C}-\bar{C}_b}$.

Now, using the fact that, if $0 \leq X \leq a + b\sqrt{X}$, then $X \leq a + b^2$ (Lemma 38 from [18]), we can obtain the bound
$$\sum_{k,h,s,a} w_h^{\pi_k}(s,a;P)|\langle(P_h-P_{h,k})(.|s,a),(V_{\bar{r}_k,h+1}^{\pi_k}(\cdot;P_k)-V_{r,h+1}^{\pi_k}(\cdot;P))\rangle|$$

$$\leq \frac{4H^5 S^2 AL}{\bar{C}-\bar{C}_b} + \frac{4S^2 H^{4.5}\sqrt{AL}}{\bar{C}-\bar{C}_b}\left(\sum_k (V_r^{\pi_k}(s_1;P)-V_{\bar{r}_k}^{\pi_k}(s_1;P_k))\right)^{1/2} + \frac{S^4 H^9 AL}{(\bar{C}-\bar{C}_b)^2}. \quad (38)$$

Substituting the above bound and (34) in (33), we obtain,
$$\sum_k V_{\bar{r}_k}^{\pi_k}(P) - V_{\bar{r}_k}^{\pi_k}(P_k) \leq$$

$$S\sqrt{AH^4 K} + \frac{H^5 S^2 AL}{\bar{C}-\bar{C}_b} + \frac{S^2 H^{4.5}\sqrt{AL}}{\bar{C}-\bar{C}_b}\left(\sum_k (V_r^{\pi_k}(s_1;P)-V_{\bar{r}_k}^{\pi_k}(s_1;P_k))\right)^{1/2} + \frac{S^4 H^9 AL}{(\bar{C}-\bar{C}_b)^2}.$$

Using the above bound in (32), we obtain,

$$\sum_k V_r^{\pi_k}(P) - V_{\bar{r}_k}^{\pi_k}(P_k) \leq S\sqrt{AH^4K} + \frac{H^5 S^2 AL}{\bar{C} - \bar{C}_b}$$

$$+ \frac{S^2 H^{4.5}\sqrt{A}L}{\bar{C} - \bar{C}_b}\left(\sum_k (V_r^{\pi_k}(s_1; P) - V_{\bar{r}_k}^{\pi_k}(s_1; P_k))\right)^{1/2} + \frac{S^4 H^9 AL}{(\bar{C} - \bar{C}_b)^2} + \frac{S\sqrt{AH^6 K}}{\bar{C} - \bar{C}_b}. \quad (39)$$

The left hand side of the above inequality is non-negative, since $V_{\bar{r}_k}^{\pi_k}(P_k) \leq V_r^{\pi^*}(P)$, from lemma 18, and $V_r^{\pi^*}(P) \leq V_r^{\pi_k}(P)$, since $\pi^*$ is the optimal policy on $P$. This equation is again of the form $0 \leq X \leq a + b\sqrt{X}$, where $X = \sum_k V_r^{\pi_k}(P) - V_{\bar{r}_k}^{\pi_k}(P_k)$. Using the same result we used to get (38), we deduce that $X \leq a + b^2$, and hence,

$$\sum_k V_r^{\pi_k}(P) - V_{\bar{r}_k}^{\pi_k}(P_k) \leq \frac{S\sqrt{AH^6 K}}{\bar{C} - \bar{C}_b} + S\sqrt{AH^4 K} + \frac{H^5 S^2 AL}{\bar{C} - \bar{C}_b} + \frac{S^4 H^9 AL}{(\bar{C} - \bar{C}_b)^2} + \frac{S^4 H^9 AL}{(\bar{C} - \bar{C}_b)^2}$$

$$\leq \tilde{O}(\frac{H}{\bar{C} - \bar{C}_b} S\sqrt{AH^4 K}).$$

and hence, from (32), $\sum_k V_r^{\pi_k}(P) - V_r^{\pi^*}(P) \leq \tilde{O}(\frac{H}{\bar{C} - \bar{C}_b}S\sqrt{AH^4 K'})$.

Moreover, from proposition (5), we have that $\pi_k \in \Pi_{\text{safe}}$ for all $k \in [K]$, with probability $1 - 5\delta$. $\quad \square$

**Lemma 19** (Bonus optimism). *For any $(s, a, h, k)$, conditioned on good event, we have that*
$$r_h(s_h, a_h) - \bar{r}_{h,k}(s_h, a_h) + \left(< P_h(.|s_h, a_h) - P_{h,k}(.|s_1, a_1), V_{r,h+1}^{\pi_k}(.; P) >\right) \geq 0, \text{ and, for any}$$
$\pi, s, h, k$, *it holds that* $V_{\bar{r}_k, h}^{\pi}(s; P_k) \leq V_{r,h}^{\pi}(s; P)$.

*Proof.* Consider
$$\bar{r}_{h,k}(s_h, a_h) - r_h(s_h, a_h) + < P_{h,k}(.|s_h, a_h) - P_h(.|s_h, a_h), V_{r,h+1}^{\pi_k}(.; P) >$$

$$\leq \sum_{s'} |(P_{h,k} - P_h)(s'|s, a)||V_{r,h+1}^{\pi_k}(s')| - \frac{H}{\bar{C} - \bar{C}_b}\beta_{h,k}^r(s_h, a_h) - \frac{H^2}{\bar{C} - \bar{C}_b}\bar{\beta}_{h,k}^p(s_1, a_1)$$

$$\leq H\bar{\beta}_{h,k}^p(s_h, a_h) - \frac{H^2}{\bar{C} - \bar{C}_b}\bar{\beta}_{h,k}^p(s_h, a_h) - \frac{H}{\bar{C} - \bar{C}_b}\beta_{h,k}^r(s_h, a_h)$$

$$\leq 0,$$

where the last inequality is due to the fact that $\bar{C} - \bar{C}_b \leq H$.

Similarly, by value difference lemma 16, we have,
$V_{\bar{r}_k, h}^{\pi}(s; P_k) - V_{r,h}^{\pi}(s; P)$

$$= \mathbb{E}\left[\sum_{h'=h}^H \bar{r}_{h',k}(s_{h'}, a_{h'}) - r_{h'}(s_{h'}, a_{h'}) - (P_{h'} - P_{h',k})(.|s_{h'}, a_{h'})V_{h'+1,r}^{\pi}(.; P)|s_h = s, \pi, P_k\right]$$

$$\leq 0,$$

where the last inequality is obtained just earlier. $\quad \square$

**Lemma 20.** *We prove inequality $(e)$ in bounding term $(i)$, i.e., we prove that,*
$$\sum_{h,s,a} w_{h+1}^{\pi_k}(s, a, P)(V_{r,h+1}^{\pi_k}(s; P) - V_{\bar{r}_k, h+1}^{\pi_k}(s; P_k)) \leq H(V_r^{\pi_k}(s_1; P) - V_{\bar{r}_k}^{\pi_k}(s_1; P_k))$$

$$+ H\sum_{h,s,a} w_h^{\pi_k}(s, a, P)| < (P_h - P_{h,k})(.|s, a), (V_{\bar{r}_k, h+1}^{\pi_k}(\cdot; P_k) - V_{r,h+1}^{\pi_k}(\cdot; P)) > |.$$

*Proof.* Let us start with,
$V_{r,1}^{\pi_k}(s_1; P) - V_{\bar{r}_k,1}^{\pi_k}(s_1; P_k)$
$$= \mathbb{E}\left[V_{r,1}^{\pi_k}(s_1; P) - r_1(s_1, a_1) - \langle P_1(.|s_1, a_1), V_{\bar{r}_k,2}^{\pi_k}(.; P_k)\rangle|\pi_k, P\right] +$$
$$+ \mathbb{E}\left[r_1(s_1, a_1) + \langle P_1(.|s_1, a_1), V_{\bar{r}_k,2}^{\pi_k}(.; P_k) - V_{\bar{r}_k,1}^{\pi_k}(s_1; P_k)\rangle|\pi_k, P\right]$$
$$= \mathbb{E}\left[\langle P_1(.|s_1, a_1), (V_{r,2}^{\pi_k}(.; P) - V_{\bar{r}_k,2}^{\pi_k}(.; P_k))\rangle|\pi_k, P\right]$$
$$+ \mathbb{E}\left[r_1(s_1, a_1) + \langle P_1(.|s_1, a_1)V_{\bar{r}_k,2}^{\pi_k}(.; P_k)\rangle - \bar{r}_{1,k}(s_1, a_1) - \langle P_{1,k}(.|s_1, a_1)V_{\bar{r}_k,2}^{\pi_k}(s_1; P_k)\rangle|\pi_k, P\right]$$

$$= \mathbb{E}\left[\langle P_1(.|s_1, a_1), (V_{r,2}^{\pi_k}(.; P) - V_{\bar{r}_k, 2}^{\pi_k}(.; P_k))\rangle | P, \pi_k\right] +$$
$$\mathbb{E}\left[\langle (P_1 - P_{1,k})(.|s_1, a_1), (V_{\bar{r}_k, 2}^{\pi_k}(.; P_k) - V_{r,2}^{\pi_k}(.; P))\rangle\right] +$$
$$\mathbb{E}\left[r_1(s_1, a_1) - \bar{r}_{1,k}(s_1, a_1) + \langle (P_1 - P_{1,k})(.|s_1, a_1), V_{r,2}^{\pi_k}(s_1; P)\rangle | \pi_k, P\right]$$
$$\overset{(a)}{\geq} \mathbb{E}\left[V_{r,2}^{\pi_k}(.; P) - V_{\bar{r}_k, 2}^{\pi_k}(.; P_k) | s_1, P, \pi_k\right] +$$
$$\mathbb{E}\left[\langle (P_1 - P_{1,k})(.|s_1, a_1), V_{\bar{r}_k, 2}^{\pi_k}(.; P_k) - V_{r,2}^{\pi_k}(.; P)\rangle | \pi_k, P\right],$$

where inequality $(a)$ is from Lemma 19. Iterating this relation, for $h$ times, we get,
$$V_{r,1}^{\pi_k}(P) - V_{\bar{r}_k, 1}^{\pi_k}(P_k)$$
$$\geq \mathbb{E}\left[V_{r,h}^{\pi_k}(.; P) - V_{\bar{r}_k, h}^{\pi_k}(.; P_k) | s_1, P, \pi_k\right] +$$
$$\sum_{h'=1}^{h-1} \mathbb{E}\left[< (P_{h'} - P_{h',k})(.|s_{h'}, a_{h'}), V_{\bar{r}_k, h'+1}^{\pi_k}(.; P_k) - V_{r,h'+1}^{\pi_k}(.; P) | s_1, \pi_k, P\right].$$

Summing this relation for $h \in \{2, \ldots, H\}$, we get,
$$H\left[V_{r,1}^{\pi_k}(P) - V_{\bar{r}_k, 1}^{\pi_k}(P_k)\right] -$$
$$\sum_{h=2}^{H}\sum_{h'=1}^{h-1} \mathbb{E}\left[< (P_{h'} - P_{h',k})(.|s_{h'}, a_{h'}), V_{\bar{r}_k, h'+1}^{\pi_k}(.; P_k) - V_{r,h'+1}^{\pi_k}(.; P) > | s_1, \pi_k, P\right]$$
$$\geq \sum_{h=2}^{H} \mathbb{E}\left[V_{r,h}^{\pi_k}(.; P) - V_{\bar{r}_k, h}^{\pi_k}(.; P_k) | s_1, P, \pi_k\right].$$

Hence, we obtain,
$$\sum_{h,s,a} w_{h+1}^{\pi_k}(s, a, P)(V_{r,h+1}^{\pi_k}(s; P) - V_{\bar{r}_k, h+1}^{\pi_k}(s; P_k)) = \sum_{h=2}^{H} \mathbb{E}\left[V_{r,h}^{\pi_k}(.; P) - V_{\bar{r}_k, h}^{\pi_k}(.; P_k) | s_1, P, \pi_k\right]$$
$$\leq H\left[V_{r,1}^{\pi_k}(P) - V_{\bar{r}_k, 1}^{\pi_k}(P_k)\right]$$
$$+ \sum_{h=2}^{H}\sum_{h'=1}^{h-1} \mathbb{E}\left[- < (P_{h'} - P_{h',k})(.|s_{h'}, a_{h'}), V_{\bar{r}_k, h'+1}^{\pi_k}(.; P_k) - V_{r,h'+1}^{\pi_k}(.; P) > | s_1, \pi_k, P\right]$$
$$\leq H\left[V_{r,1}^{\pi_k}(P) - V_{\bar{r}_k, 1}^{\pi_k}(P_k)\right]$$
$$+ \sum_{h=2}^{H}\sum_{h'=1}^{H} \mathbb{E}\left[| < (P_{h'} - P_{h',k})(.|s_{h'}, a_{h'}), V_{\bar{r}_k, h'+1}^{\pi_k}(.; P_k) - V_{r,h'+1}^{\pi_k}(.; P) > | | s_1, \pi, P\right]$$
$$\leq H\left[V_{r,1}^{\pi_k}(P) - V_{\bar{r}_k, 1}^{\pi_k}(P_k)\right]$$
$$+ H\sum_{h=1}^{H} \mathbb{E}\left[| < (P_h - P_{h,k})(.|s_h, a_h), V_{\bar{r}_k, h+1}^{\pi_k}(.; P_k) - V_{r,h+1}^{\pi_k}(.; P) > | | s_1, \pi_k, P\right].$$
$$\square$$

## E  Detailed Description of Experiment Environments and Algorithm Implementation

### E.1  Experiment Environments

**Factored CMDP environment:** The factored CMDP is represented in Fig. 3. The state space is $\mathcal{S} = \{1, 2, 3\}$, and the action space is $\mathcal{A} = \{1, 2\}$, where action 1 corresponds to moving one step to the right and 2 corresponds to staying put. Objective cost is $r(s, 1) = 0, \forall s \in \mathcal{S}$, and $r(s, 2) = s, \forall s \in \mathcal{S}$. The constraint cost is, $c(s, 1) = 0, \forall s \in \mathcal{S}$, and $c(s, 2) = 1, \forall s \in \mathcal{S}$. The probability transition matrix under action 1 is, $\begin{pmatrix} 0 & 1 & 0 \\ 0 & 0 & 1 \\ 1 & 0 & 0 \end{pmatrix}$, and under action 2, it is $\begin{pmatrix} 1 & 0 & 0 \\ 0 & 1 & 0 \\ 0 & 0 & 1 \end{pmatrix}$.

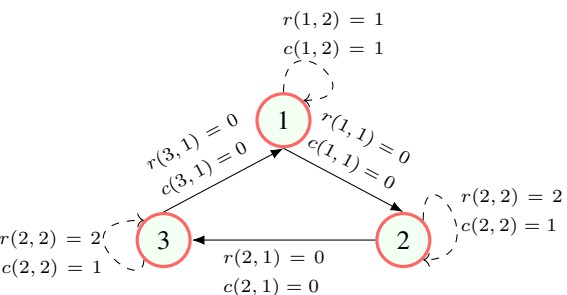

Figure 3: Illustrating the Factored CMDP environment. Environment transitions to next state with action 1, and stays put with action 2.

**Media Streaming Environment:** Here, we model the media streaming control from a wireless base station. The base station provides two types of service to a device, a fast service and a slow service. The packets received are stored in a media buffer at the device. The goal is to minimize the cost of having an empty buffer (which may result in stalling of the video), while keeping the utilization of fast service below certain level.

We denote by $A_h$, the number of incoming packets into the buffer, and by $B_h$, the number of packets leaving the buffer. The state of the environment, denoted as $s_h$ in $h^{th}$ step, is the media buffer length. It evolves as, $s_{h+1} = \min\{\max(0, s_h + A_h - B_h), N\}$. We consider $N = 20$ as the maximum buffer length in our experiment. The action space is $\mathcal{A} = \{1, 2\}$, i.e., the action is to use either fast server 1 or slow server 2. We assume that the service rates of the servers have independent Bernoulli distributions, with parameters $\mu_1 = 0.9$, and $\mu_2 = 0.1$, where $\mu_1$ corresponds to the fast service. The media playback at the device is also Bernoulli with parameter $\gamma$. Hence, $A_h$ is a random variable with mean either $\mu_1$ or $\mu_2$ depending on the action taken, and $B_h$ is a random variable with mean $\gamma$. These components constitute the unknown transition dynamics of our environment.

The objective cost is $r(s, a) = \mathbb{1}\{s = 0\}$. i.e., it has a value of 1, when the buffer hits zero, and is zero everywhere else. Our constraint cost is $c(s, a) = \mathbb{1}\{a = 1\}$, i.e., there is a constraint cost of 1 when the fast service is used, and is zero otherwise. We then constrain the expected number of times the fast service is used to $\bar{C} = H/2$, in a horizon of length $H = 10$.

**Inventory Control Environment:** We consider a single product inventory control problem [7]. Our environment evolves according to a finite horizon CMDP, with horizon length $H = 7$, where each time step $h \in [H]$ represents a day of the week. In this problem, our goal is to maximize the expected total revenue over a week, while keeping the expected total costs in that week below a certain level. We do not backlog the demands.

The storage has a maximum capacity $N = 6$, which means it can store a maximum of 6 items. We denote by $s_h$, the state of the environment, as the amount of inventory available at $h^{th}$ day. The action $a_h$ is the amount of inventory the agent purchases such that the inventory does not overflow. Thus, the action space $\mathcal{A}_s \in \{0, \ldots, N - s\}$, for the state $s$. The exogenous demand is represented by $d_h$, which is a random variable representing the stochastic demand for the inventory on the $h^{th}$ day. We assume $d_h$ to be in $\{0, \cdots, N\}$ with distribution $[0.3, 0.2, 0.2, 0.05, 0.05]$. If the demand is higher than the inventory and supply, the excess demand will not be met. The state evolution then follows as $s_{h+1} = \max\{0, s_h + a_h - d_h\}$.

We define the rewards and costs as follows. The revenue is generated as, $f(s, a, s') = 8(s + a - s')$, when $s' > 0$, and is 0 otherwise. The reward obtained in state $(s, a)$ is then the expected revenue over all next states $s'$, $r(s, a) = \mathbb{E}[f(s, a, s')]$. The cost associated with the inventory has two components. Firstly, there is a purchase cost when the inventory is brought in, which is a fixed cost of 4 units, plus a variable cost of $2a$, which increases with the amount of purchase. Secondly, we also have a non decreasing holding cost $s$, for storing the inventory. Hence, the cost in $(s, a)$ is $c(s, a) = 4 + 2a + s$. We normalize the rewards and costs to be in the range $[0, 1]$. Our goal is to maximize the expected total revenue over a week ($H = 7$), while keeping the expected total costs in that week below a threshold $\bar{C} = H/2$.

## E.2 Details of the Implementation

We now describe the algorithms described in the introduction.

**AlwaysSafe:** AlwaysSafe [36] shows empirical results that only depict the expected cost of various policies deduced from their linear programs versus the optimal expected cost. We note that this comparison is not a reasonable measure of regret, since cumulative regret can be linear even though the expected costs are close.

We consider a factored CMDP environment described previously. For implementing this algorithm, in each episode, we solve the LP4 linear program described in [36], based on the observations. For solving LP4, one also needs an abstract CMDP as described in Section 3 of [36]. We follow their description to construct such a model for the factored CMDP. The confidence intervals for AlwaysSafe algorithm are same as the ones for OptCMDP algorithm from [18] and of DOPE. We notice that the regret of Always safe algorithms indeed grow at a linear rate.

**OptPessLP:** We implement Algorithm 1 from [28]. We choose the baseline policy by solving the corresponding CMDP with a more conservative constraint. For the factored CMDP and Media Control environment, we choose the constraint as $0.1\bar{C}$, and solve the MDP to obtain $\pi_b$ and the corresponding cost $\bar{C}_b$. Similarly, for the inventory control environment, we choose the constraint as $0.2\bar{C}$. These choices of $\pi_b$ are the same for both OptPessLP and DOPE, for a fair comparison. We play $\pi_b$ when the condition in Equation 9 from [28] is met, otherwise we choose the maximizing policy from their linear program. We choose the confidence intervals as specified in their work, without any scaling. Despite the results suggested by theory, we notice that its empirical performance is very poor in every environment we consider.

**OptCMDP:** We implement Algorithm 1 from [18]. This algorithm solves the linear program that minimizes the objective cost with optimism in the model. Since this algorithm does not consider zero-violation setting, we expect to see constraint violation of the same order as the regret. We use the confidence intervals as specified in their work, without any scaling.

**DOPE:** We implement Algorithm 1 from our work. The choice of baseline policy $\pi_b$ is exactly the same as that for OptPessLP for each environment. $\pi_b$ is played until $K_0$ episodes, as provided by Proposition 4. Then, the algorithm solves the linear program given by Equation (10) to obtain $\pi_k$ in episode $k$.

The details for the linear program formulations are given in Appendices A and B. DOPE, AlwaysSafe and OptCMDP algorithms use the Extended LP formulations, while OptPessLP uses the regular LP formulation.

For each environment, each of these algorithms are run for 20 random seeds, and are averaged to obtain the regret plots in the figures.

## E.3 Experiment Results for Inventory Control Environment

We show the performance of our DOPE algorithm in Inventory Control Environment. As before, we compare it against the OptCMDP Algorithm 1 in [18], and and OptPess-LP algorithm from [29]. Also, we choose the optimal policy from a conservative constrained problem (with a stricter constraint) as the baseline policy. We use $\bar{C}_b = 0.1\bar{C}$.

Fig. 4(a) compares the objective regret for the inventory control environment incurred by each algorithm with respect to the number of episodes. As we see in this figure, in the initial episodes, the objective regret of DOPE grows linearly with number of episodes. Later, the growth rate of regret changes to square root of number of episodes. We see that this change of behavior happens after $K_0$ episodes specified by Proposition 4. Hence, the linear growth rate indeed corresponds to the duration of time in which the base policy is employed. In conclusion, the regret for DOPE algorithm depicted in Figures 4(a) matches the result of Theorem 3. Next, the OptPess-LP algorithm performs quite badly in terms of objective regret, as it fails to achieve $\sqrt{K}$ regret performance within the chosen number of episodes. It thus shows the same issue of excessive pessimism observed in the other environments. Finally, we observe that the objective regret of OptCMDP is lower than DOPE. This behavior can be attributed to the fact that in order to perform safe exploration, DOPE includes a pessimistic penalty in the constraint (8).

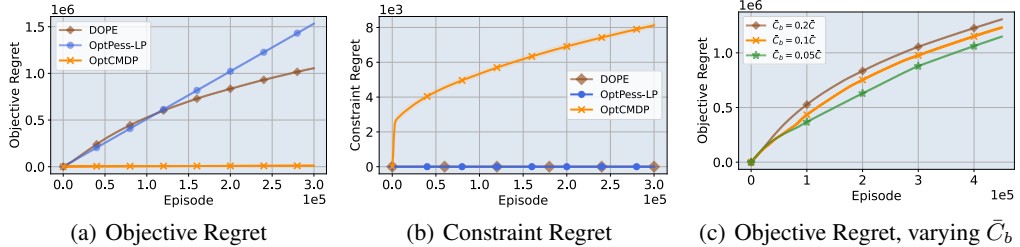

(a) Objective Regret       (b) Constraint Regret       (c) Objective Regret, varying $\bar{C}_b$

Figure 4: Illustrating the Objective Regret and Constraint Regret for the Inventory Control Environment.

Fig. 4(b) compares the regret in constraint violation for DOPE, OptPess-LP and OptCMDP algorithms for the inventory control setting. Here, we see that DOPE and OptPess-LP do not violate the constraint, while OptCMDP incurs a regret that grows sublinearly. This figure shows that DOPE does indeed perform safe exploration as proved, while OptCMDP violates the constraints during learning.

Finally, Fig. (4(c)) compares the optimality regret for various baseline policies. Again, the takeaway here is that a good baseline policy is helpful, although the variation across different baseline policies is not very large.