# OpenReview forum: "DOPE: Doubly Optimistic and Pessimistic Exploration for Safe Reinforcement Learning"
_NeurIPS.cc/2022/Conference — NeurIPS 2022 Accept_

### Official Review · Reviewer_ucst · 2022-07-01

**Rating:** 8
**Confidence:** 4
**Soundness:** 4 excellent
**Presentation:** 4 excellent
**Contribution:** 3 good

**Summary:**

Existing safe reinforcement learning (RL) algorithms are basically optimistic w.r.t. reward (via exploration bonus) and pessimistic w.r.t. constraints. Such algorithms are excessively pessimistic; thus, empirical performance is bad due to a long learning process. The authors present Doubly Optimistic and Pessimistic
Exploration (DOPE) algorithm for safe RL, which is doubly optimistic w.r.t. reward and model while being pessimistic w.r.t safety. The authors theoretically analyze the objective regret and constraint violation, which is better than OptPessLP algorithm. Finally, they empirically show that DOPE performs better than baselines in multiple benchmark tasks.

**Questions:**

[Q1] Can the authors provide me with an intuitive explanation of which of the innovations in this paper led to the reduction of objective regret? I think I understand the DOPE algorithm and followed the proof, but I do not fully understand why the DOPE algorithm can achieve smaller regret.

**Limitations:**

Limitations and potential negative societal impact have been fully addressed in lines 351-355 and lines 487-500, respectively.

**Strengths And Weaknesses:**

I think this paper is a solid paper, which will potentially contribute to the future safe RL researches.

[Strength]
- This paper is well-organized and easy to follow.
- Theoretical results are good. Especially, it is an important result that objective regret is less than OptPess-LP while constraint regret is zero.
- Empirical experiments. I deeply appreciate that the authors conducted experiments in three tasks. As the authors argue, many safe RL algorithms do not perform well in empirical experiments in spite of its beautiful theoretical properties. I personally think that DOPE algorithm is impressive in that it is backed by theory and practice.

[Weakness]
- (Note that I think this is a minor weakness) Benchmark tasks are still easy and simple.
- Source code is messy. I would like the authors to refactor it before open-sourcing it.

---

> ### Author Response · Authors · 2022-08-02
> **Response to Official Review of Paper11419 by Reviewer ucst**
>
> Thank you very much for your thoughtful comments and appreciating the contribution of our paper. We are encouraged to know that the reviewer found  our DOPE algorithm is impressive in that it is backed by theory and practice. We address  the main comments  below.
>
> **Q1.**  *"Can the authors provide me with an intuitive explanation of which of the innovations in this paper led to the reduction of objective regret? I think I understand the DOPE algorithm and followed the proof, but I do not fully understand why the DOPE algorithm can achieve smaller regret.''*
>
> **Response:**  The DOPE algorithm is doubly optimistic in that it enables optimistic exploration both in terms of the model and the objective (reward), to balance with the pessimism of the constraint.  The additional optimism enables more exploration after the initial $K_o$ episodes of applying the base policy, and so enables fast policy improvement. In turn, this faster policy improvement results in a smaller cumulative regret.  We employ the empirical Bernstein inequalities in the regret analysis of DOPE to handle the regret engendered by the additional model optimism.  The use of this approach enables us to obtain tighter bounds regret bounds, resulting in a smaller regret scaling in terms of $S$.
>
> **Q2.**  *"Source code is messy. I would like the authors to refactor it before open-sourcing it.''*:
>
> **Response:** Thank you for pointing that out.  We would indeed like to open-source the codebase and will definitely refactor it and add comments and annotations on its usage before releasing it.  Our aim is to make it available as a general purpose tool to train and evaluate model-based RL algorithms that involve solving linear programs.

---

> > ### Comment · Reviewer_ucst · 2022-08-03
> > **Thank you for the response**
> >
> > Thank you for the response. The authors' comments (especially for Q1) resolved my concerns. I do not have further questions.

---

### Official Review · Reviewer_sxzS · 2022-07-11

**Rating:** 7
**Confidence:** 2
**Soundness:** 3 good
**Presentation:** 3 good
**Contribution:** 3 good

**Summary:**

The authors propose a safe exploration algorithm that relies on the idea of both optimism and pessimism under uncertainty. The problem is formulated as a constrained MDP that is solved using a Linear Programming approach which iteratively approximates the transition model and the optimal policy, the cost, and constraints for each state-action pair.
In order to ensure that the policy is safe through this approximation, an additional term is added to the constraint objectives (pessimism) and the cost objective is relaxed with additional terms (optimism) which ensures the exploration of under-explored states. These terms are O(1/n(s,a)) and can cause the initial problem to be infeasible. Therefore,
the algorithm relies on an initial safe exploration policy that ensures that constraints are not violated. The authors demonstrate that under certain conditions, constraints are satisfied with high probability in the original problem i.e. under the true transition model. The result is an algorithm that has 0 constraint regret and that, compared to SOTA, reduces the objective regret
by a factor of O^{\tilde}(\sqrt(|S|)) where |S| is the size of the state space.

**Questions:**

Line 179: Shouldn't there be a summation over the timestep h? or it should be the number of times the pair (s,a) has been observed at timestep h before the beginning of episode k
Line 188: The summation should be until k-1 and r_{h} should be replaced by r_{h, k^{\prime}} I suppose
Line 189: What is r^{\tilde} in terms of (r^{\tilde}_{h})

**Limitations:**

The algorithm is limited to tabular settings but this is something that the authors have already pointed out.

**Strengths And Weaknesses:**

Strengths:

An algorithm that combines both advantages of pessimism and optimism approaches to safety yielding 0 constraints regret and a reduced objective regret by a factor of O^{\tilde}(\sqrt(|S|))
An analysis of the algorithm which clarifies the motivation behind the algorithm
Empirical results that demonstrate the superiority of the algorithm over SOTA
Weaknesses:

The algorithm relies on the assumption that we have an initial safe exploration policy which ensures that the constraints are always satisfied.

---

> ### Author Response · Authors · 2022-08-02
> **Response to Official Review of Paper11419 by Reviewer sxzS**
>
> We thank the reviewer for appreciating the theoretical contribution of our paper and the empirical demonstration of the proposed algorithm. We address  the main comments  below.
>
> **Q1.** $(i)$ *Line 179: Shouldn't there be a summation over the timestep $h$? or it should be the number of times the pair $(s,a)$ has been observed at timestep $h$ before the beginning of episode $k$*.
>
> $(ii)$ *Line 188: The summation should be until $k-1$*
>
> $(iii)$ *What is $\tilde{r}$ in terms of $\tilde{r}_{h}$?*
>
> **Response:** $(i)$ It is indeed the  number of times the pair $(s,a)$ has been observed at time step $h$ before the beginning of episode $k$. So, there is no summation over $h$. We  have now added a line in our paper to explicitly mention this. Thank you very much for pointing this out.
>
> $(ii)$ Thank you for pointing this out this typo. We have fixed it now.
>
> $(iii)$ We use $\tilde{c}$ and $\tilde{r}$ as the vector notation as
>  $\tilde{c}=(\tilde{c}_1\dots\tilde{c}_H) , \tilde{r} = (\tilde{r}_1\dots\tilde{r}_H)$. We have included this clearly in the revised draft now.

---

> > ### Comment · Reviewer_sxzS · 2022-08-08
> > **After Rebuttal**
> >
> > Thank you very much for the reply!

---

### Official Review · Reviewer_UkP1 · 2022-07-11

**Rating:** 4
**Confidence:** 3
**Soundness:** 2 fair
**Presentation:** 2 fair
**Contribution:** 2 fair

**Summary:**

This paper proposes an RL algorithm to learn a policy in a constrained setting without violating the safety constraints.
It relies on an initial baseline policy to collect data for the first episodes.
Eventually, it switches to a policy optimistic with respect to the transition function and reward.
To ensure the new policy is safe the paper proposes to penalize the cost function inversely to the number of times, being pessimistic with respect to the cost function.

The paper provides regret bounds for the main objective tighter than the bounds from previous algorithms that also do not violate the safety constraints (OptPess-LP).

**Questions:**


1. The current definition of regret is related to a weak measure as described in [17], which allows for "error cancellations". Considering Proposition 5, which states the policy applied in each episode is safe with high-probability, can we conclude the bound on the safety regret of DOPE is a strong measure?

2. How DOPE behaves in the Factored CMDP given a baseline with cost bound $\bar{C}_b = 0$?

3. Empirically, it seems the effect of the baseline policy in the regret seems to be contradictory to what would be expected from the regret bounds. That is, lower values of $\bar{C}_b$ would lead to lower regrets. Could you comment on this phenomenon?

### references

[17] Yonathan Efroni, Shie Mannor, and Matteo Pirotta. Exploration-exploitation in constrained MDPs. arXiv preprint arXiv:2003.02189, 2020.

**Limitations:**


Yes, the paper makes clear statements regarding the assumptions required for the proposed algorithm, namely, access to a safe initial baseline policy.

**Strengths And Weaknesses:**


### strengths

- The main novelty of the proposed algorithm compared to the previous OptPess-LP is to be optimistic with respect to the transition function in addition to optimism with respect to the reward. To compensate for the double optimism the paper also proposes a new penalty for the cost function, increasing the pessimism with respect to the cost.
- The significance of the paper seems limited to showing that the algorithm proposed has smaller regret compared to the OptPess-LP algorithm.

### weaknesses

- The empirical results could be improved considerably. Although the paper describes the AlwaysSafe-$\pi_\alpha$ algorithm, it tests the AlwaysSafe-$\pi_T$ algorithm. In [34], it is already shown that the $\pi_T$ variant can be overly conservative, it may even fail to converge to the optimal policy. An appropriate comparison should use the $\pi_\alpha$ algorithm, which can converge to the optimal policy.
- The clarity of the paper could be improved, for instance, by clarifying the connections between the different theoretical claims in Section 4.

### minor comments

- line 100: peratins -> pertains
- line 210: something is strange with "less observed less"


### references

[34] Thiago D Simão, Nils Jansen, and Matthijs TJ Spaan. AlwaysSafe: Reinforcement learning without safety constraint violations during training. 2021.

---

> ### Author Response · Authors · 2022-08-02
> **Response to Official Review of Paper11419 by Reviewer UkP1**
>
> We thank the reviewer for their thoughtful comments on our paper.   We address  the main comments  below.
>
> **Q1.**  *``The empirical results could be improved considerably. Although the paper describes the AlwaysSafe-$\pi_{\alpha}$ algorithm [34], it tests the AlwaysSafe-$\pi_T$ algorithm. In [34], it is already shown that the $\pi_T$ variant can be overly conservative, it may even fail to converge to the optimal policy. An appropriate comparison should use the $\pi_{\alpha}$ algorithm, which can converge to the optimal policy.*
>
> **Response:**
> We have now simulated both variants of AlwaysSafe in the specific factored MDP setting to illustrate their performance. The $\pi_{\alpha}$ variant mentioned by the reviewer is more complex than both OptPessLP and DOPE but we managed to complete the simulations in time.   However, we found $\pi_{\alpha}$ has a linear growth in safe objective regret in the same manner as $\pi_T,$ although the growth rate of regret is lower than the $\pi_T$ variant.  So we have no substantial changes to report on the performance of AlwaysSafe (both versions) as compared to OptPessLP or DOPE.
>
> We note that none of the versions of AlwaysSafe are proven to be objective regret optimal in the AlwaysSafe paper.  So the reviewer's remark  ``An appropriate comparison should use the $\pi_{\alpha}$ algorithm, which can converge to the optimal policy" is not accurate in our context.  We have now empirically verified the sub-optimality in terms of safe objective regret of both variants of AlwaysSafe.
>
> Note also that since neither variant of AlwaysSafe is applicable in  the general CMDP case, we do not provide a comparison in the media streaming environment. We have updated the Figures 1 (a) and (b) to include the performance curve of $\pi_{\alpha}$ algorithm, and uploaded the revised manuscript.  We hope that the reviewer will now acknowledge that we have fairly illustrated all the variants of AlwaysSafe.
>
> **Q2.**.  *``The current definition of regret is related to a weak measure as described in [17], which allows for error cancellations. Considering Proposition 5, which states the policy applied in each episode is safe with high-probability, can we conclude the bound on the safety regret of DOPE is a strong measure?''*
>
> **Response:** Our definition of safe regret is indeed a stronger measure of regret for safe RL problem compared to that of  [17]. In [17], the proposed algorithms may violate constraints during the learning phase. Hence, some policies that the algorithm chooses may outperform the optimal policy and accrue negative regret, and hence, it allows for error cancellations. Our algorithm does not  violate the constraint at all (with high probability). So, the issue of error cancellation does not arise in our case.
>
> **Q3.**.  *``Empirically, it seems the effect of the baseline policy in the regret seems to be contradictory to what would be expected from the regret bounds. That is, lower values of $\bar{C}_b$  would lead to lower regrets. Could you comment on this phenomenon?''*
>
> **Response:** As the reviewer mentions, though the regret is indeed sublinear for different values $\bar{C}_b$, the order of the exact values exhibit variations across different environments. In Fig.1(c) (factored CMDP) and Fig.2.(c) (media streaming CMDP), the regret for lower  $\bar{C}_b$ is slightly larger than that of the higher $\bar{C}_b$. In Fig.4(c) (inventory control CMDP, see Appendix E.3),  the regret for lower  $\bar{C}_b$ is slightly smaller than that of the higher $\bar{C}_b$.  We believe that that this may be due to the variations in the random exploration policies selected by the algorithm and specific nature of the environment on which is deployed.

---

> > ### Comment · Reviewer_UkP1 · 2022-08-07
> > **Thanks to the authors**
> >
> > I would like to thank the authors for addressing my concerns. In particular, for taking the time to run the experiments with the adaptive AlwaysSafe algorithm.
> >
> > It would be great if the authors could comment on my second question:
> > - How the DOPE algorithm behaves given a baseline with cost bound zero?

---

> > > ### Author Response · Authors · 2022-08-08
> > > **Response to Reviewer UkP1's comment**
> > >
> > > We apologize for not explicitly responding to that question in our response.  We had tried to immediately address those questions that we thought were of greatest importance to the reviewer.  To answer this question, please note that DOPE can operate with any baseline policy, including those that do not visit every state-action pair.  Hence, even when $\bar{C}_b=0,$  which means that the baseline policy in the factored MDP never uses action 2 (see Figure 3 in the appendix), DOPE will still explore sufficiently to learn the optimal policy.  We have run DOPE in a variety of cases, including those in which the baseline policy is similarly restricted and so we are justified in this claim both analytically and empirically. For completeness, we ran the simulation and have included the case of $\bar{C}_b=0,$ in Figure 1(c) in the revised draft, which shows performance on expected lines.
> > >
> > > Thank you for all the insightful comments and questions. We hope that we have addressed all of the reviewer's questions satisfactorily and do hope that the reviewer rating can be adjusted to reflect our effort in doing so.

---

### Meta-Review · Area_Chair_qS18 · 2022-08-26

**Recommendation:** Accept
**Confidence:** Less certain

**Metareview:**

I went through the paper, reviews and responses. This is a borderline paper with reasonable theoretical analysis but weak experimental results. Lack of comparisons to practical safe RL algorithms is not an advantage.
I tend to accept. I'm also ok with a borderline reject.

**Award:**

No

---

### Decision · Program_Chairs · 2022-09-14

Accept